# Ethanol abolishes vigilance-dependent astroglia network activation in mice by inhibiting norepinephrine release

Liang Ye [1,7], Murat Orynbayev[1,7], Xiangyu Zhu [1,2], Eunice Y. Lim[1,3], Ram R. Dereddi[4], Amit Agarwal [4,5], Dwight E. Bergles [5,6], Manzoor A. Bhat [1,3] & Martin Paukert [1,3✉]

Norepinephrine adjusts sensory processing in cortical networks and gates plasticity enabling adaptive behavior. The actions of norepinephrine are profoundly altered by recreational drugs like ethanol, but the consequences of these changes on distinct targets such as astrocytes, which exhibit norepinephrine-dependent $Ca^{2+}$ elevations during vigilance, are not well understood. Using in vivo two-photon imaging, we show that locomotion-induced $Ca^{2+}$ elevations in mouse astroglia are profoundly inhibited by ethanol, an effect that can be reversed by enhancing norepinephrine release. Vigilance-dependent astroglial activation is abolished by deletion of $\alpha_{1A}$-adrenergic receptor from astroglia, indicating that norepinephrine acts directly on these ubiquitous glial cells. Ethanol reduces vigilance-dependent $Ca^{2+}$ transients in noradrenergic terminals, but has little effect on astroglial responsiveness to norepinephrine, suggesting that ethanol suppresses their activation by inhibiting norepinephrine release. Since abolition of astroglia $Ca^{2+}$ activation does not affect motor coordination, global suppression of astroglial networks may contribute to the cognitive effects of alcohol intoxication.

[1] Department of Cellular and Integrative Physiology, University of Texas Health Science Center at San Antonio, San Antonio, TX, USA. [2] Xiangya School of Medicine, Central South University, Changsha, Hunan, China. [3] Center for Biomedical Neuroscience, University of Texas Health Science Center at San Antonio, San Antonio, TX, USA. [4] The Chica and Heinz Schaller Research Group, Institute for Anatomy and Cell Biology, Heidelberg University, Heidelberg, Germany. [5] Solomon H. Snyder Department of Neuroscience, Johns Hopkins University School of Medicine, Baltimore, MD, USA. [6] Johns Hopkins Kavli Neuroscience Discovery Institute, Baltimore, MD, USA. [7] These authors contributed equally: Liang Ye, Murat Orynbayev. ✉email: paukertm@uthscsa.edu

Among noradrenergic nuclei, locus coeruleus (LC) is the main source of norepinephrine (NE) in the mammalian brain with projections to almost all areas of the forebrain and cerebellum[1–3]. NE signaling plays key roles in adjusting neural activity during different behavioral states, such as arousal, attention, reward, motivation, and stress[4,5]. A powerful example of this occurs in primary visual cortex (V1), in which the onset of locomotion is associated with NE-dependent modulation of visual processing gain, enhancing the sensitivity of cortical neurons to incoming visual information[6]. The numerous adrenergic receptor subtypes, their expression by many distinct cell types and the diffuse manner in which NE is released have posed challenges for understanding how global modulation is accomplished. Therefore, the molecular mechanisms that are responsible for these state changes have not been completely established.

Voluntary and enforced locomotion induces global and coordinated $Ca^{2+}$ activation in cerebellar Bergmann glia (BG) and in cortical astrocytes that is dependent on $\alpha_1$-adrenergic receptors[7–9], despite an abundance in other neurotransmitter receptors leading to intracellular $Ca^{2+}$ release in astroglia in culture or slice preparations[10]. These signals can also be evoked by sensory and aversive stimuli[9,11], a feature that is anatomically supported by the multitude of inputs to the LC from cortex, amygdala, and cerebellar Purkinje cells[2]. Genetic, opto-, and chemogenetic manipulations of astroglia intracellular $Ca^{2+}$ release can affect cortical plasticity, hippocampus-dependent learning and attention[12–14], and raise the possibility that vigilance-dependent astroglia $Ca^{2+}$ elevations play a role in cognitive brain function. It is currently not known which $\alpha_1$-adrenergic receptor subtype mediates vigilance-dependent astroglia network activation, whether NE acts directly on astroglia receptors or how this widespread $Ca^{2+}$ signal is regulated.

Modulation of the LC–NE system has immediate consequences for the level of behavioral activity and vigilance. Activation of noradrenergic neurons in LC opposes the action of general anesthetics while their inhibition potentiates general anesthesia[15]. During wakefulness, an inverted U-shaped dose–effect relationship between brain NE and cognitive performance has been found suggesting a precise control of LC activity[16]. Ethanol, the most prevalent recreational drug, can cause alterations of LC activity[17,18], suggesting that it could also affect vigilance-dependent astroglia $Ca^{2+}$ activation. Acute ethanol exposure already at low dosage can lead to impairment of vigilant attention and cognitive performance[19–21] and chronic ethanol exposure can lead to morphological changes in the central nervous system characteristic for cognitive decline[22]. Yet, the mechanisms underlying the effect of ethanol on vigilant attention and cognitive performance are not well understood. In anesthetized rats, ethanol impairs the reliability of foot shock stimulation-induced excitation of unidentified neurons in LC, suggesting less synchronized release of NE, while the mean firing rate is not affected[23]. Similarly, electrophysiological recordings in acute brain slices revealed that ethanol lowers the spontaneous firing rate of unidentified neurons in LC, suggesting that acute ethanol exposure might partially suppress NE release[24]. However, the LC also contains local interneurons[25] and it is not known if ethanol affects regional, phasic NE release, and subsequent activation of adrenergic receptors on cellular targets when animals are not anesthetized. Microdialysis has been used to measure brain NE levels during exposure to ethanol in awake animals[26–28]; however, the time resolution did not allow to detect phasic NE release which plays a role in encoding salience information[29].

Ethanol affects $Ca^{2+}$ dynamics in cultured astrocytes. Astroglia in resting mice, in the slice preparation and in culture exhibit spontaneous, locally restricted microdomain $Ca^{2+}$ transients that involve the mitochondrial permeability transition pore and are facilitated by reactive oxygen species (ROS)[30–32]. Ethanol increases spontaneous, locally restricted $Ca^{2+}$ elevations in cultured astrocytes that involves ROS production[33–35]. As a consequence, ethanol increases glutamate release from cultured astrocytes and stimulates the expression of glial fibrillary acidic protein, a marker of reactive astrocytes. In addition, it has been reported that high concentrations of ethanol can inhibit muscarinic $Ca^{2+}$ responses in cultured astrocytes[36]. Together these findings suggest that ethanol can have acute as well as long-lasting effects on astroglia biology; however, little is known about the effects of ethanol on vigilance-dependent, noradrenergic responses in astroglia in awake behaving animals.

Here we used in vivo two-photon (2P) imaging to determine whether acute ethanol intoxication alters vigilance-dependent activation of astroglial networks, and whether disruption of adrenergic signaling in astroglia impacts the ability of ethanol to impair motor coordination. We used locomotion as a natural stimulus to elicit phasic LC activation and transgenic mice expressing genetically encoded $Ca^{2+}$ indicators (GECIs) specifically in cerebellar BG, cortical astrocytes or noradrenergic neurons and their terminals, in combination with 2P microscopy to monitor $Ca^{2+}$ dynamics in awake head-restrained mice. We found that vigilance-dependent activation of astroglia depends on direct activation of $\alpha_{1A}$-adrenergic receptors in these cells. Activation of these glial networks during enhanced arousal was strongly suppressed by moderate levels of ethanol. This suppression was caused by an inhibition of NE release, rather than a decrease in sensitivity of astroglia to NE. By comparing the time course of ethanol inhibition of astroglial activation and motor coordination, we conclude that loss of vigilance-dependent BG $Ca^{2+}$ elevations is not responsible for ataxic motor behavior.

## Results

**Ethanol impairs vigilance-dependent cerebellar BG $Ca^{2+}$ activation.** To investigate the mechanisms of BG activation in vivo we used $Slc1a3\text{-}CreER^T;R26\text{-}lsl\text{-}GCaMP3$ mice that expressed the genetically encoded $Ca^{2+}$ indicator GCaMP3 in all BG[7], installed chronic cranial windows above lobulus simplex/crus I of the cerebellar hemisphere, habituated the head-restrained mice to a linear treadmill and employed 2P microscopy (920 nm) to relate BG $Ca^{2+}$ dynamics to locomotion activity (Fig. 1a). Here, we use the term "BG activation" synonymously with "vigilance-dependent" or "locomotion-induced" BG $Ca^{2+}$ elevation. We imaged cross-sections of BG processes within the molecular layer approximately 60 μm from the pial surface (Fig. 1b). Since it is not possible to ascribe individual processes to a particular BG cell in this configuration, we defined 64 equally sized regions of interest (ROIs) in a checkerboard style to assess the spatial activity pattern within the BG population (Fig. 1b, c). To elicit consistent BG activation, we applied short episodes (5 s) of enforced locomotion (80–110 mm/s) (Fig. 1c, green bars), a strategy that induces astroglia $Ca^{2+}$ elevations with the same time course, refractory period and pharmacological profile as voluntary locomotion[7]. Intraperitoneal (i.p.) injection of ethanol (2 g/ kg body weight) resulted in a rapid, almost complete inhibition of BG activation (Fig. 1c). Ethanol inhibition of BG activation was dose-dependent, reaching almost complete inhibition at 1.5 g/kg (i.p.). This ethanol dosage is within the range used for acute ethanol exposure studies of rodents[23,28,37], and caused temporary ataxic motor coordination as discussed below, but did not impair their ability to walk on the treadmill while head restrained (Fig. 1d). Ethanol reduced the mean $Ca^{2+}$ elevation (Fig. 1f) and slowed the rise of the residual $Ca^{2+}$ elevation (Fig. 1g). BG activation was highly temporally coordinated, resulting in considerable correlation among ROIs. With increasing dosage,

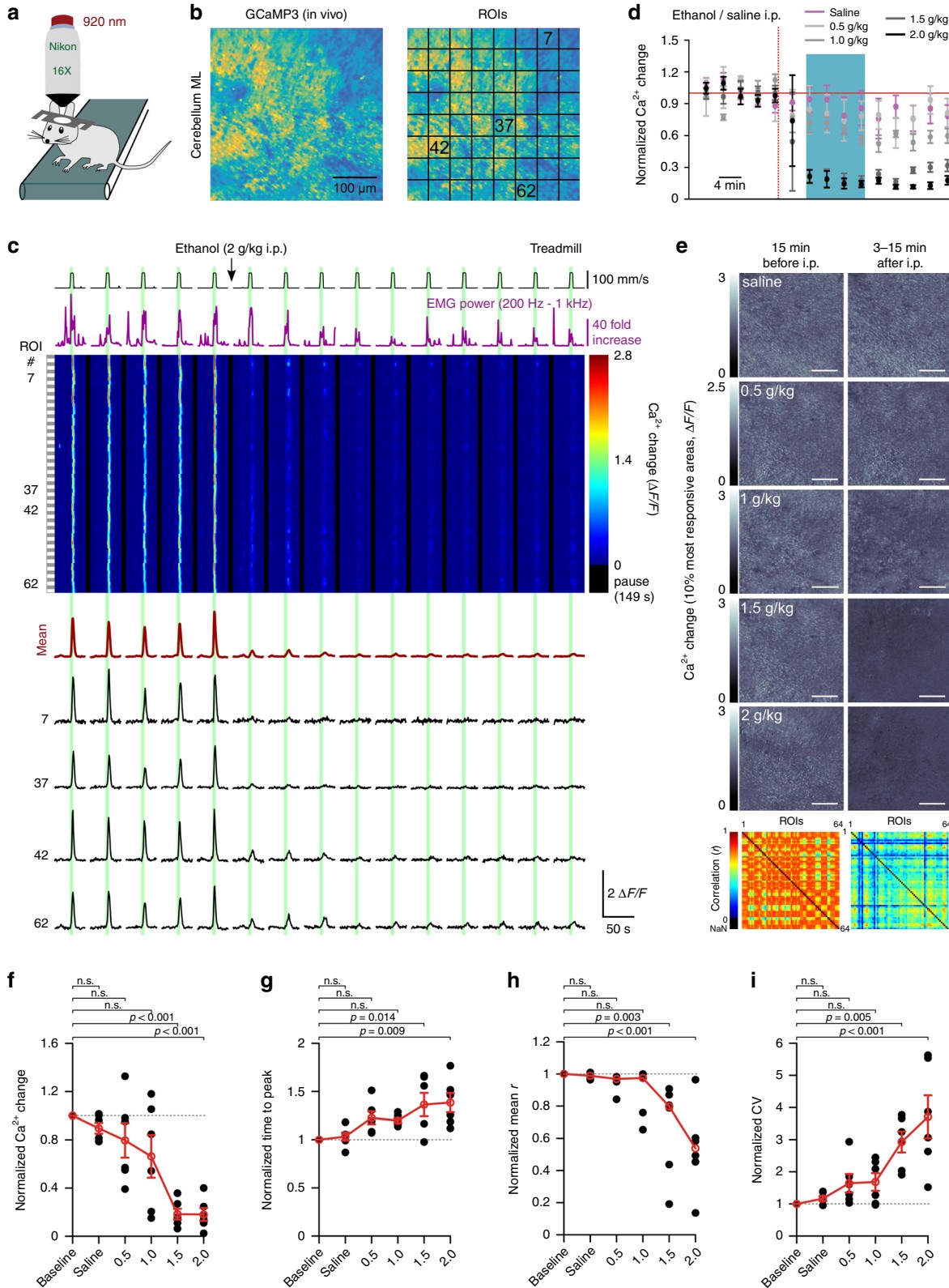

ethanol disrupted the coordination of BG activation (Fig. 1e), resulting in reduced correlation (Fig. 1e, h) and increased Ca²⁺ response variability among ROIs (Fig. 1i). The inhibition by 2.0 g/kg i.p. ethanol was substantial for at least 45 min (Fig. 2a–c) and was completely reversible within 6 h (Fig. 2d–g). Ethanol inhibited BG activation also in the cerebellar vermis, irrespective of whether it was triggered by locomotion or aversive air puff

stimulation to the whisker pad (Supplementary Fig. 1). These findings are consistent with the possibility that ethanol impairs NE signaling to all cerebellar BG, thereby suppressing vigilance-dependent Ca²⁺ elevations.

**α$_{1A}$-adrenergic receptor is required for vigilance-dependent BG Ca²⁺ activation.** To understand how ethanol inhibits astroglia

**Fig. 1 Ethanol impairs vigilance-dependent cerebellar BG Ca$^{2+}$ activation. a** Scheme of 2P Ca$^{2+}$ imaging in awake, head-restrained mouse on a motorized linear treadmill. **b** Left, Pseudocoloured in vivo Ca$^{2+}$ image of BG processes, tangential optical section—cerebellum molecular layer, *Slc1a3-CreER$^T$;R26-lsl-GCaMP3* mouse. Right, Locations of regions of interest (ROIs) used in **c**. **c** Upper, Pseudocolour plot of all ROIs' Ca$^{2+}$ responses. Green bars, enforced locomotion. Lower, Corresponding average Ca$^{2+}$ response trace of all ROIs (dark red) and traces representing numbered ROIs. **d** Time course of effect of saline/ethanol injection (red dotted line) on $\Delta F/F_{10s}$ (normalized mean Ca$^{2+}$ change (($F - F_{median}$ of baseline)/$F_{median}$ of baseline[7]) within 10 s from onset of locomotion). Normalized to average first five trials (baseline). Blue bar, effect analysis time window. mean ± SEM, $n = 6$ mice per dosage. **e** Upper, Maximum response plots during indicated experimental episodes. Scale bar, 100 μm. Lower, Linear Pearson correlation coefficient plots between individual ROIs' Ca$^{2+}$ change traces before and after 2 g/kg ethanol. Numbers of independent experimental repetitions with similar results were: 6 (saline), 7 (0.5 g/kg), 6 (1.0 g/kg), 6 (1.5 g/kg), and 6 (2.0 g/kg). **f-i** Population data, mean normalized values within blue bar in **d**. Numbers under abscissa, g/kg i. p. ethanol. Data represent: mean $\Delta F/F_{10s}$ (**f**), mean time to peak (baseline time from onset of locomotion to peak of population response; **g**), mean correlation coefficient (baseline Pearson linear correlation coefficient among Ca$^{2+}$ change traces within 20 s from onset of locomotion; $r$; **h**) and coefficient of variation (CV) among $\Delta F/F$ traces of individual ROIs (**i**). Red symbols, mean ± SEM if data follow Gaussian distribution; without error bars, red symbols represent median if data do not follow Gaussian distribution. Lines between dots support readability. $n = 6$ mice per dosage; one-way ANOVA (**f** $F_{(5, 30)} = 11.110$, $p < 0.001$; **g** $F_{(5, 30)} = 4.963$, $p = 0.002$ and **i** $F_{(5, 30)} = 9.543$, $p < 0.001$) or Kruskal–Wallis test (**h**) were followed by Tukey–Kramer correction; n.s. not significant. Source data are provided as a Source Data file.

activation, whether through a mechanism upstream of, at or downstream of noradrenergic receptors, we first need to understand how NE causes BG activation. Pharmacological studies revealed the necessity of α$_1$-adrenergic receptors for BG (Supplementary Fig. 2) and cortical astrocyte activation[7,9]. However, the cellular location and molecular identity of the α$_1$-adrenergic receptor are not known. Among the three α$_1$-adrenergic receptor subtypes, α$_{1A}$ RNA has consistently been detected in astrocytes[38,39]. Indeed, global knockout of α$_{1A}$-adrenergic receptors[40] in *Slc1a3-CreER$^T$;Ai95* mice resulted in almost complete loss of BG activation (Fig. 3a–c). Residual Ca$^{2+}$ elevations had slower kinetics (Fig. 3d) and occurred less reliably, resulting in a much-reduced correlation among ROIs (Fig. 3e). Together, these results indicate that vigilance-dependent BG Ca$^{2+}$ activation results from the activation of α$_{1A}$-adrenergic receptors.

**Activation of α$_{1A}$-adrenergic receptors on BG causes vigilance-dependent Ca$^{2+}$ activation.** To explore the possibility that direct α$_{1A}$-adrenergic receptor activation on BG was the mechanism underlying Ca$^{2+}$ activation, we generated mice in which this receptor was selectively deleted from astroglia (Fig. 4a and Supplementary Fig. 3) and investigated the effect on BG activation using two different Cre driver mouse lines. In an attempt to maximize cellular specificity of α$_{1A}$-adrenergic receptor knockout we identified a Cre driver mouse line for BG-specific recombination within the cerebellum. The abundance of zinc-finger protein family members in studies that aimed to identify BG-enriched genes not previously known to be expressed in glia[41] led us to search the GENSAT atlas for further related candidate genes that lack expression in neurons. Among these, Gli1 expression is glia-specific and the GENSAT expression analysis indicates preferential abundance in the adult cerebellar molecular layer. Indeed, tamoxifen induction during the fourth postnatal week in *Gli1-CreER$^{T2}$* mice[42] crossed with Ai95 transgenic mice revealed exclusive GCaMP6f expression within the cerebellum in BG (Fig. 4a, left). Of note, granular layer velate astrocytes identified by expression of the Ca$^{2+}$-binding protein S100β (Fig. 4a, left) lack *Gli1-CreER$^{T2}$*-driven expression of GCaMP6f in contrast to abundant expression when driven by *Aldh1l1-CreER$^{T2}$* (ref. [11]), which allows tamoxifen-inducible Cre recombination in granular layer velate astrocytes throughout life (Fig. 4a, right). *Gli1-CreER$^{T2}$*-mediated BG-specific knockout of α$_{1A}$-adrenergic receptors resulted in a considerable, but incomplete reduction in BG activation (Fig. 4b). In view of the almost complete loss of BG activation in global Adra1a$^{-/-}$ mice (Fig. 3) the incomplete reduction of BG activation in the *Adra1a$^{cKO/cKO}$* mice could be due either to incomplete recombination of the two floxed *Adra1a* alleles among BGs or to a prolonged stability of α$_{1A}$-adrenergic

receptors in the BG plasma membrane following gene deletion; both scenarios should lead to a broad range of variability in locomotion-induced Ca$^{2+}$ responsiveness within the BG population. To test this hypothesis, we investigated the responsiveness of individual BG cells to NE compared to adenosine triphosphate (ATP) in acute cerebellar slices (Fig. 4c). We superfused slices with NE followed by 15 min washout and then ATP to obtain a Gq protein-coupled receptor-dependent reference response[43] and to confirm that BG in these slices were capable of exhibiting intracellular Ca$^{2+}$ release. BG Ca$^{2+}$ dynamics were monitored with 2P microscopy (920 nm) with as many BG somata in focus as possible. In *Adra1a$^{wt/wt}$* slices, NE as well as ATP induced reliable monophasic or oscillatory Ca$^{2+}$ elevations in almost all BG (Fig. 4c). The second response (ATP) was in 57% of BG weaker than the first response (NE), represented as bright signal in the NE−ATP response difference plot and by the inset cumulative probability plot in the lower left panel of Fig. 4c, leaving only 43% of *Adra1a$^{wt/wt}$* BG respond less to NE. In contrast, in *Adra1a$^{cKO/cKO}$* slices, 91% of BG responded less to NE than to the subsequent application of ATP (dark signal in the NE−ATP response difference plot and inset cumulative probability plot (Fig. 4c). Consistent with these findings, driving Cre recombination in BGs as well as velate astrocytes with the *Aldh1l1-CreER$^{T2}$* line yielded similar results (Fig. 4a and Supplementary Fig. 4). To further investigate location of α$_{1A}$-adrenergic receptor activation in the cerebellum and recombination efficiency in *Aldh1l1-CreER$^{T2}$;Adra1a$^{cKO/cKO}$* mice we applied NE to acute cerebellar slices and found that no intracellular Ca$^{2+}$ release occurred in two of the most abundant cerebellar cell populations, Purkinje cells and granule cells (Supplementary Fig. 5a, b), and Cre recombination in BG and velate astrocytes was sufficient to eliminate the vast majority of *Adra1a* mRNA in whole cerebellum from *Aldh1l1-CreER$^{T2}$;Adra1a$^{cKO/cKO}$* mice (Supplementary Fig. 5c, d). These findings, together with the almost complete loss of BG responsiveness with global knockout of α$_{1A}$-adrenergic receptors (Fig. 3), strongly suggest that α$_{1A}$-adrenergic receptor is almost exclusively expressed on astroglia in the cerebellum and that its direct activation on BG is required for vigilance-dependent BG Ca$^{2+}$ activation.

**Ethanol does not impair the responsiveness of BG to NE.** As a promiscuous drug[44] ethanol could alter the responsiveness of BGs to NE, or inhibit BG intracellular Ca$^{2+}$ release directly. To address whether ethanol inhibition of BG activation was due to direct inhibition of α$_{1A}$-adrenergic receptors on BG or suppression of intracellular downstream signaling in BG, or rather due to an alteration of the NE concentration reaching α$_{1A}$-adrenergic receptors on BG during locomotion, we investigated the effect of

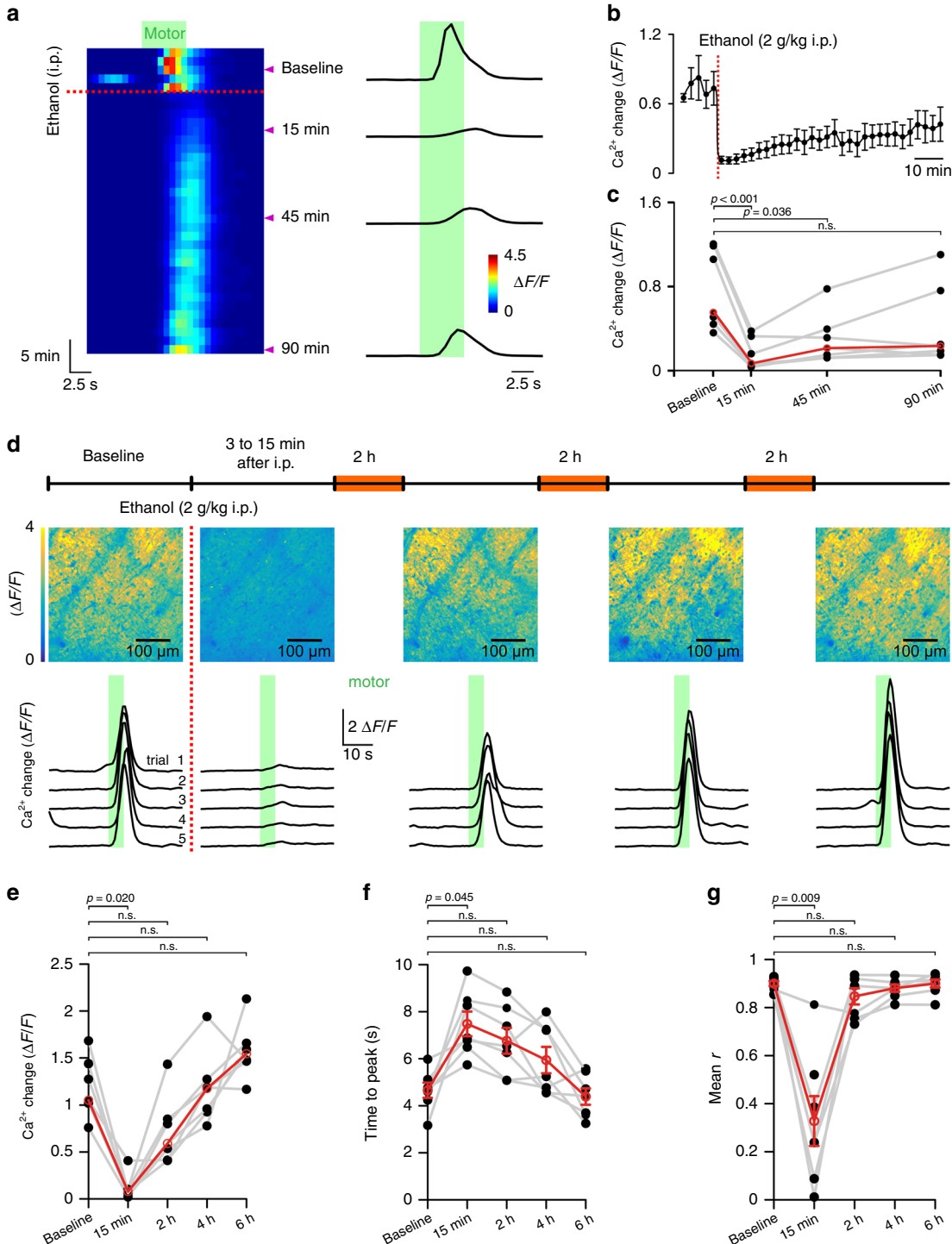

**Fig. 2 Vigilance-dependent BG Ca²⁺ activation recovers from ethanol inhibition within 6 h. a** Left, Pseudocolour plot of time course of Ca²⁺ responses of BG processes of *Aldh1l1-CreER^T2; R26-lsl-GCaMP6f(Ai95)* mouse. Each row represents average Ca²⁺ response to one trial of enforced locomotion (green bar) of all ROIs determined as in Fig. 1. Ca²⁺ responses to consecutive locomotion trials are vertically concatenated. Red dotted line indicates injection of 2 g/kg i.p. ethanol. Right, Representative Ca²⁺ response traces of the trials indicated by purple arrowheads. **b** Time course of mean ± SEM $\Delta F/F_{10s}$ values from seven mice. **c** Ca²⁺ responses at indicated time points following ethanol injection with baseline representing the average of five trials. Red symbols indicate median from seven mice. n.s. not significant; Friedman test followed by Tukey–Kramer correction. **d** Upper, Maximum response plots of BG processes at indicated time points (seven independent experimental repetitions with similar results). Lower, Staggered overlay of Ca²⁺ response traces at 200 s intervals within the respective time window. **e–g** Population data: mean $\Delta F/F_{10s}$ (**e**), mean time to peak (**f**), and mean correlation coefficient (*r*) (**g**) within indicated time windows. Red symbols indicate median (**e**) or mean ± SEM (**f, g**) from seven mice. Friedman test (**e**) and repeated measures ANOVA (**f**, $F(4, 24) = 10.606$, $p < 0.001$ and **g** $F(4, 24) = 26.949$, $p < 0.001$) were followed by Tukey–Kramer correction; n.s. not significant. Source data are provided as a Source Data file.

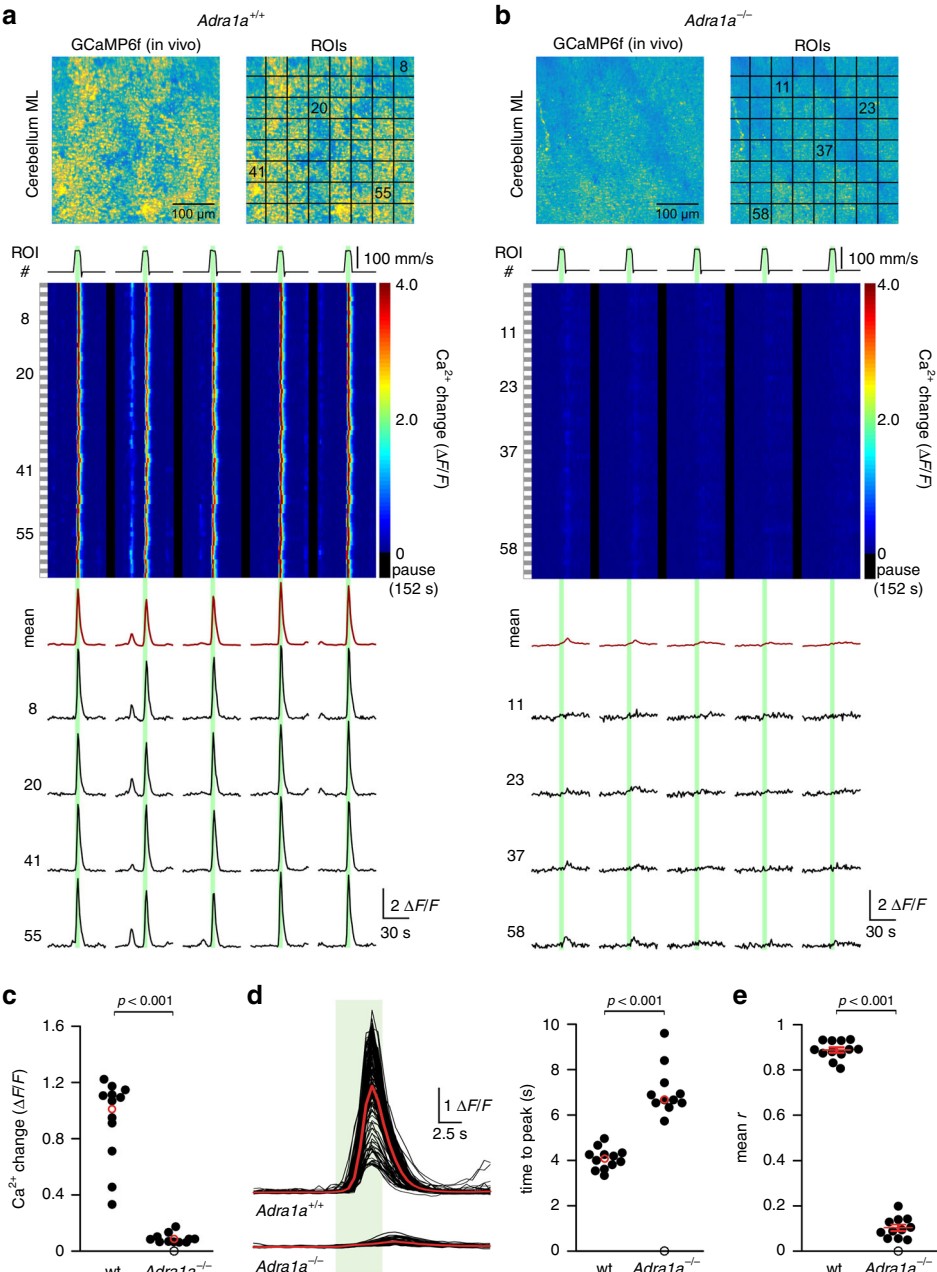

**Fig. 3 $\alpha_{1A}$-adrenergic receptor is required for vigilance-dependent BG Ca$^{2+}$ activation. a** Upper, Pseudocoloured fluorescence images of BG processes of *Slc1a3-CreER$^T$;Ai95; Adra1a$^{+/+}$* mouse and ROI assignments (12 independent experimental repetitions with similar results). Middle, Pseudocolour plot of ROIs' Ca$^{2+}$ dynamics. Green bars, enforced locomotion. Lower, Average Ca$^{2+}$ response of all ROIs (dark red) and Ca$^{2+}$ response traces within number-defined ROIs. **b** Same as **a**; however, for *Slc1a3-CreER$^T$;Ai95;Adra1a$^{-/-}$* mouse (11 independent experimental repetitions with similar results). **c** Population data: mean enforced locomotion-induced Ca$^{2+}$ activation (5 s before peak to 5 s after peak) of *Adra1a$^{+/+}$* (wt) and *Adra1a$^{-/-}$* mice. Red symbols indicate median. Empty circle indicates that one *Adra1a$^{-/-}$* mouse had no detectable Ca$^{2+}$ response. $n = 12$ mice, respectively; Kruskal–Wallis test. **d** Left, Overlay of peak-aligned *Adra1a$^{+/+}$* and *Adra1a$^{-/-}$* fluorescence traces. Green bar, enforced locomotion. Red traces represent median traces. Right, Population data: time from onset of locomotion to peak amplitude. Red symbol indicates median. $n = 12$ mice, respectively; Kruskal–Wallis test. **e** Population data: mean correlation coefficient ($r$) (10 s before peak to 10 s after peak); red symbols indicate mean ± SEM. $n = 12$ mice, respectively; $t(21) = 43.717$; unpaired, two-tailed Student's $t$-test. Source data are provided as a Source Data file.

ethanol on NE-induced BG Ca$^{2+}$ elevations in acute cerebellar slices. Acute parasagittal cerebellar slices were prepared from 4-week-old *Aldh1l1-CreER$^{T2}$;Ai95* mice. BG soma Ca$^{2+}$ dynamics were monitored with 2P imaging (920 nm) of GCaMP6f fluorescence. First, ATP was bath applied to obtain a reference response, avoiding desensitization of $\alpha_{1A}$-adrenergic receptors. Following washout, NE was bath applied in the absence (Fig. 5a) or presence (Fig. 5b) of 30 mM ethanol, a concentration reached

in plasma following intraperitoneal injection of 2.0 g/kg ethanol in the mouse strains used in this study (see below), and expected to exceed the concentration of ethanol reached in central nervous system extracellular space following intraperitoneal injection of 1.5 g/kg ethanol in mice[45], a dosage that was sufficient to almost completely inhibit BG activation (Fig. 1). However, ethanol did not inhibit the peak NE-induced Ca$^{2+}$ elevation (Fig. 5c). These findings demonstrate that ethanol interaction with $\alpha_{1A}$-adrenergic

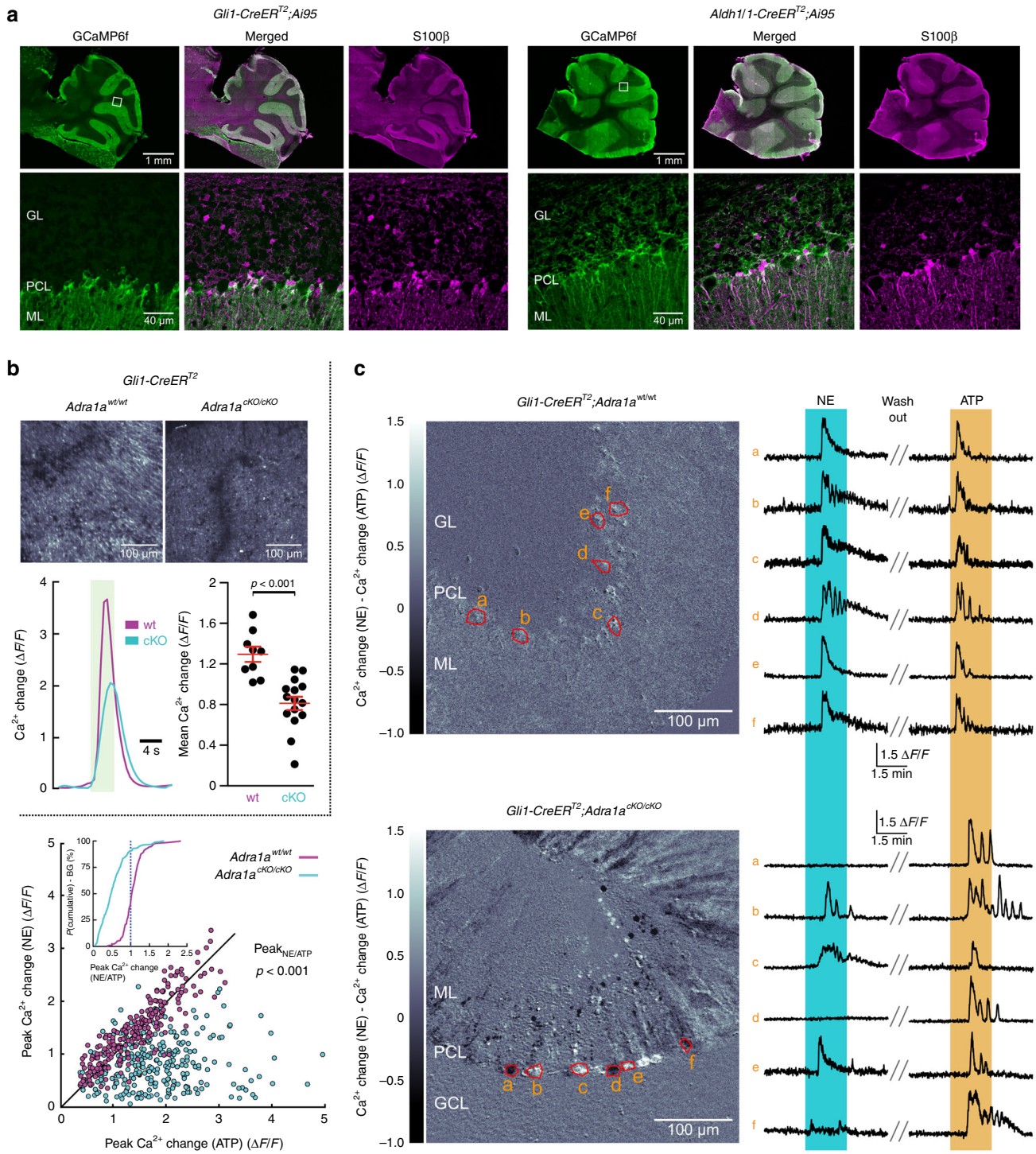

receptors on BG or any downstream target cannot account for almost complete loss of BG activation by ethanol. Rather, ethanol inhibition of noradrenergic neurons in LC or their local terminals may diminish locomotion-induced NE release and cause a failure of activation of BG $\alpha_{1A}$-adrenergic receptors.

**Ethanol impairs vigilance-dependent primary visual cortex astrocyte Ca²⁺ activation.** If ethanol inhibited the activation of noradrenergic neurons in LC, due to the high correlation of locomotion-induced Ca²⁺ elevations in cerebellar BG and primary visual cortex (V1) astrocytes[7], we would predict that V1 astrocyte activation would also be suppressed by ethanol. To test

this hypothesis, we imaged V1 astrocytes in deep layer I, 60–80 μm from the pial surface (Fig. 6a). Image frames were averaged during peak Ca²⁺ activation to draw ROIs representing individual astrocytes and assess the activity pattern within the V1 astrocyte population (Fig. 6a, b). The same locomotion paradigm was applied as described for Fig. 1. Locomotion induced reliable astrocyte Ca²⁺ elevations (Fig. 6b) and i.p. injection of ethanol (2 g/kg body weight) resulted in rapid, almost complete inhibition of V1 astrocyte activation (Fig. 6b). Ethanol inhibition of V1 astrocyte activation, as observed for BG in the cerebellum, was similarly dose-dependent and lasted for at least 30 min (Fig. 6c). Ethanol reduced the mean Ca²⁺ elevation (Fig. 6e). V1 astrocyte

**Fig. 4 Activation of $\alpha_{1A}$-adrenergic receptors on BG causes vigilance-dependent Ca$^{2+}$ activation. a** Left, Cerebellar section from 6-week-old *Gli1-CreER$^{T2}$; Ai95* mouse immunostained for eGFP (GCaMP6f; green) and S100β (magenta). White box highlights area displayed at high magnification. Right, Same for 6-week-old *Aldh1l1-CreER$^{T2}$;Ai95* mouse. GL granular layer, PCL Purkinje cell layer, ML molecular layer. Three independent experimental repetitions were obtained with similar results. **b** Upper, Representative BG locomotion-induced Ca$^{2+}$ response plots of *Gli1-CreER$^{T2}$;Ai95;Adra1a$^{wt/wt}$* and *Gli1-CreER$^{T2}$;Ai95; Adra1a$^{cKO/cKO}$* mice. Lower left, Overlay of averaged Ca$^{2+}$ change traces of wt mice (9 fields of view (FOVs) from 3 mice) and cKO mice (15 FOVs from 5 mice). Lower right, Population data of mean Ca$^{2+}$ changes. Red symbols indicate mean ± SEM. Unpaired, two-tailed Student's *t*-test ($t(22) = 4.707$). **c** Responsiveness of BG to bath application of NE (30 μM) and ATP (100 μM), respectively, in acute cerebellar slices—2P Ca$^{2+}$ imaging of *Gli1-CreER$^{T2}$; Ai95;Adra1a$^{wt/wt}$* or *Gli1-CreER$^{T2}$;Ai95;Adra1a$^{cKO/cKO}$* mice (postnatal day (PND) 160–180). Upper left, Visualization of the difference between responses to NE and ATP of *Adra1a$^{wt/wt}$* BG. Structures that responded stronger to NE than to ATP appear brighter. Red circumferences indicate ROIs enclosing individual BG somata in the PCL. Upper right, Ca$^{2+}$ change traces for each ROI. Double-line blanks 11 min of washout with artificial cerebrospinal fluid (aCSF). Lower middle and right, Same for *Adra1a$^{cKO/cKO}$* BG in the *Gli1-CreER$^{T2}$;Ai95* background. Lower left, Population data: Comparison of peak Ca$^{2+}$ change in individual BG in response to NE or ATP. Diagonal illustrates equi-response line. Kruskal–Wallis test compared the ratio of the BG peak response to NE over the peak response to ATP (*Adra1a$^{wt/wt}$*, 229 BG cells from 13 slices from 3 mice; *Adra1a$^{cKO/cKO}$*, 282 BG cells from 15 slices from 3 mice). Inset illustrates the cumulative probability distribution of individual BGs' responses to NE normalized to their respective response to ATP. The dashed line corresponds to the equi-response line. Source data are provided as a Source Data file.

activation reached the peak more slowly than in BG (time to peak$_{V1}$, 8.5 ± 0.2 s, $n = 31$ (15 mice), time to peak$_{BG}$, 5.3 ± 0.1 s, $n = 30$ (14 mice), $t(59) = 15.784$, $p < 0.001$, unpaired, two-tailed Student's *t*-test) indicating that local signaling constraints differ[7]. Consistent with this notion, in contrast to its effect on the kinetics of BG Ca$^{2+}$ activation, ethanol did not slow down the rise of the residual V1 astrocyte activation (Fig. 6f). There was less coordination among V1 astrocytes' activation than among BG (baseline $r_{V1}$, 0.83 (0.36–0.94), $n = 31$ (15 mice); baseline $r_{BG}$, 0.89 (0.59–0.95), $n = 30$ (14 mice), $p = 0.019$, Kruskal–Wallis test), and similar to its effect on BG activation, ethanol dose-dependently disrupted the coordination of V1 astrocytes' activation resulting in reduced correlation (Fig. 6d, g) and in a trend towards increased Ca$^{2+}$ response variability among ROIs (Fig. 6h). Together, these findings reveal that vigilance-dependent astroglia Ca$^{2+}$ activation is sensitive to ethanol inhibition in multiple brain regions and in distinct cell types, consistent with a global inhibition of the LC–NE system by ethanol.

**Pharmacological facilitation of NE release restores ethanol-inhibited vigilance-dependent cerebellar BG Ca$^{2+}$ activation.** To explore the possibility that ethanol inhibits NE release and does not cause inhibition of BG activation through facilitation of BG $\alpha_{1A}$-adrenergic receptor desensitization[46], we took advantage of the presence of autoinhibitory $\alpha_2$-adrenergic receptors on NE terminals and somata, which limit excitability of NE neurons and NE release[47]. As has been found in acute brain slice experiments, inhibition of $\alpha_2$-adrenergic receptors can restore ethanol-suppressed LC neuronal spiking activity[48]. Our prediction was that if the net effect of ethanol was to reduce extracellular NE, inhibition of $\alpha_2$-adrenergic receptors could help restore astroglia activation in the presence of ethanol. We used the highest dosage of ethanol tested in Fig. 1 (2 g/kg i.p.) and observed a robust, almost complete inhibition of BG activation (Fig. 7a, b). Usually, this inhibition would last for at least 45 min (Fig. 2a–c); however, when the $\alpha_2$-adrenergic receptor antagonist MK912 was injected 15 min after ethanol injection, BG activation was rapidly and completely restored (Fig. 7b). MK912 restored BG activation dose-dependently (Fig. 7c and Supplementary Fig. 6), with 0.03 mg/kg i.p. as the lowest dose sufficient for complete restoration. MK912 completely restored BG mean Ca$^{2+}$ elevation following ethanol inhibition (Fig. 7e), the rate of onset of BG activation following a deceleration by ethanol (Fig. 7f), the correlation among ROIs of BG activation following a reduction of correlation by ethanol (Fig. 7d, g) and the CV following an increase by ethanol (Fig. 7h). These findings suggest that acute ethanol exposure reduces NE release following onset of locomotion contributing to the loss of BG Ca$^{2+}$ activation.

**Ethanol inhibition of vigilance-dependent excitation of noradrenergic neurons accounts for loss of BG Ca$^{2+}$ responsiveness.** The findings above are consistent with ethanol inhibiting locomotion-induced release of NE leading to reduced activation of $\alpha_{1A}$-adrenergic receptors on BG. However, ethanol is a promiscuous drug that could as well affect other local circuit elements that might contribute to reduced BG activation, and facilitation of NE release by inhibiting $\alpha_2$-adrenergic autoreceptors would then restore BG activation through compensatory enhancement of NE release beyond baseline levels. To shed light on these possible scenarios, we directly monitored locomotion-induced Ca$^{2+}$ dynamics in cerebellar molecular layer noradrenergic terminals. We expressed the membrane-anchored form of GCaMP6f specifically in noradrenergic neurons in dopamine β-hydroxylase (*Dbh*)-*Cre*[49];*Lck-GCaMP6f$^{flox}$* (ref. [11]) mice (Fig. 8a). Like for the analysis of BG activation, we used an unbiased checkerboard ROI scheme for analysis of temporal and spatial population activity within 50 μm × 50 μm fields of axons and terminals (Fig. 8b). Locomotion-induced coordinated transient Ca$^{2+}$ elevations throughout the FOV were rapidly and strongly inhibited by ethanol (2 g/kg i.p.) (Fig. 8c) and the inhibition persisted for at least 30 min (Supplementary Fig. 7). Ethanol reduced the mean Ca$^{2+}$ elevation (Fig. 8c, e) and slowed the rise of the residual Ca$^{2+}$ elevation (Fig. 8f). Similar to BG activation, locomotion-induced noradrenergic terminal Ca$^{2+}$ elevations were highly temporally coordinated leading to considerable correlation among ROIs, which was impaired by acute ethanol exposure (Fig. 8d, g). Remarkably, the same dosage of the $\alpha_2$-adrenergic receptor antagonist MK912 that was sufficient to completely restore BG activation (0.03 mg/kg i.p., Fig. 7) caused a fast recovery of all parameters of locomotion-induced NE terminal Ca$^{2+}$ elevations (Fig. 8d–g). These findings demonstrate that inhibition of vigilance-dependent noradrenergic terminal Ca$^{2+}$ elevations by acute exposure to ethanol is sufficient to account for the loss of BG activation. Noradrenergic terminal Ca$^{2+}$ elevations can either be attenuated by local inhibition of noradrenergic terminal excitability, or by inhibition of noradrenergic neurons in LC. Local superfusion of ethanol to the acutely exposed cerebellar hemisphere did not inhibit BG activation (Fig. 9a), confirming our finding that ethanol did not affect the responsiveness of BG to NE from acute cerebellar slice experiments (Fig. 5), and suggesting that ethanol did not affect the intrinsic excitability of noradrenergic terminals. In contrast, when we inhibited BG activation by administration of 2 g/kg i.p. ethanol, topical application of MK912 was sufficient to recover BG responses (Fig. 9b). These findings suggest that ethanol at this dosage inhibits noradrenergic neurons leading to a reduction in frequency of action

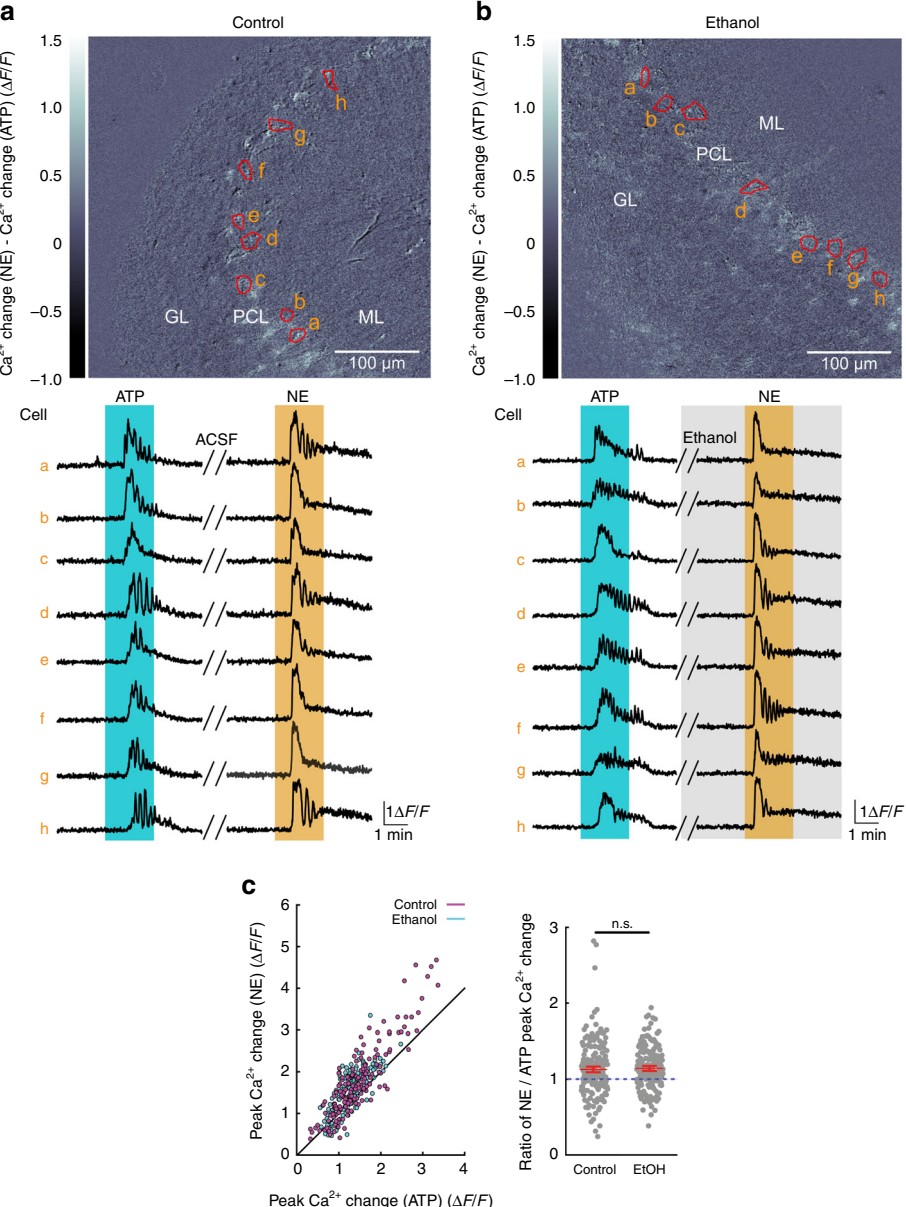

**Fig. 5 Ethanol does not impair the responsiveness of BG to NE. a** Upper, Visualization of the difference between responses to ATP (100 µM) and NE (30 µM) of BG in acute cerebellar slices from 4-week-old *Aldh1l1-CreER^T2;Ai95* mice. Red circumferences indicate ROIs enclosing representative BG somata in the PCL. Lower, $Ca^{2+}$ change traces for each ROI. Double-line blanks 11 min of washout with aCSF. Nine independent experimental repetitions were obtained with similar results. **b** Same as in **a**; however, NE was bath applied in the presence of ethanol (30 mM). Ten independent experimental repetitions were obtained with similar results. **c** Left, Population data: Comparison of peak $Ca^{2+}$ change in individual BG in response to NE or ATP. Diagonal illustrates equi-response line. Right, Ratio of $Ca^{2+}$ responses of individual BG to NE/ATP. Red symbols indicate mean ± SEM. Control, $n = 174$ BG from nine slices and three mice; ethanol, $n = 170$ BG from 10 slices and 3 mice; unpaired, two-tailed Student's *t*-test ($t(342) = -0.222$; n.s. not significant). EtOH ethanol. Source data are provided as a Source Data file.

potentials reaching the terminals, rather than to a complete abolition of action potentials.

**Ethanol inhibition of vigilance-dependent BG $Ca^{2+}$ activation is independent from its effect on motor coordination.** Acute ethanol intoxication comprises ataxic motor coordination as well as non-motor cognitive impairments[50]. To test if ethanol inhibition of BG activation contributes to ataxic motor coordination we first compared the time course of inhibition of BG activation following acute ethanol exposure (see Fig. 2) with the time course of the corresponding plasma ethanol concentrations. For all mouse strains used in this study, plasma concentrations reached a

peak within 15 min following intraperitoneal administration and declined steadily within 6 h (Fig. 10a), resembling the time course of inhibition of BG activation by ethanol (Fig. 2). We then quantified the effect of acute ethanol exposure on motor coordination using the CatWalk XT behavioral paradigm, a camera-based automated gait analysis system[51]. In all, 2 g/kg i.p. ethanol induced robust ataxic motor coordination, reflected by a slowed leg swing and a less precise positioning of paws on the track (Fig. 10b, c and Supplementary Movies 1 and 2). Remarkably, even though BG activation was still considerably inhibited 45 min following 2 g/kg i.p. ethanol (Fig. 2a–c), mice performed on the CatWalk XT similarly as observed under baseline conditions

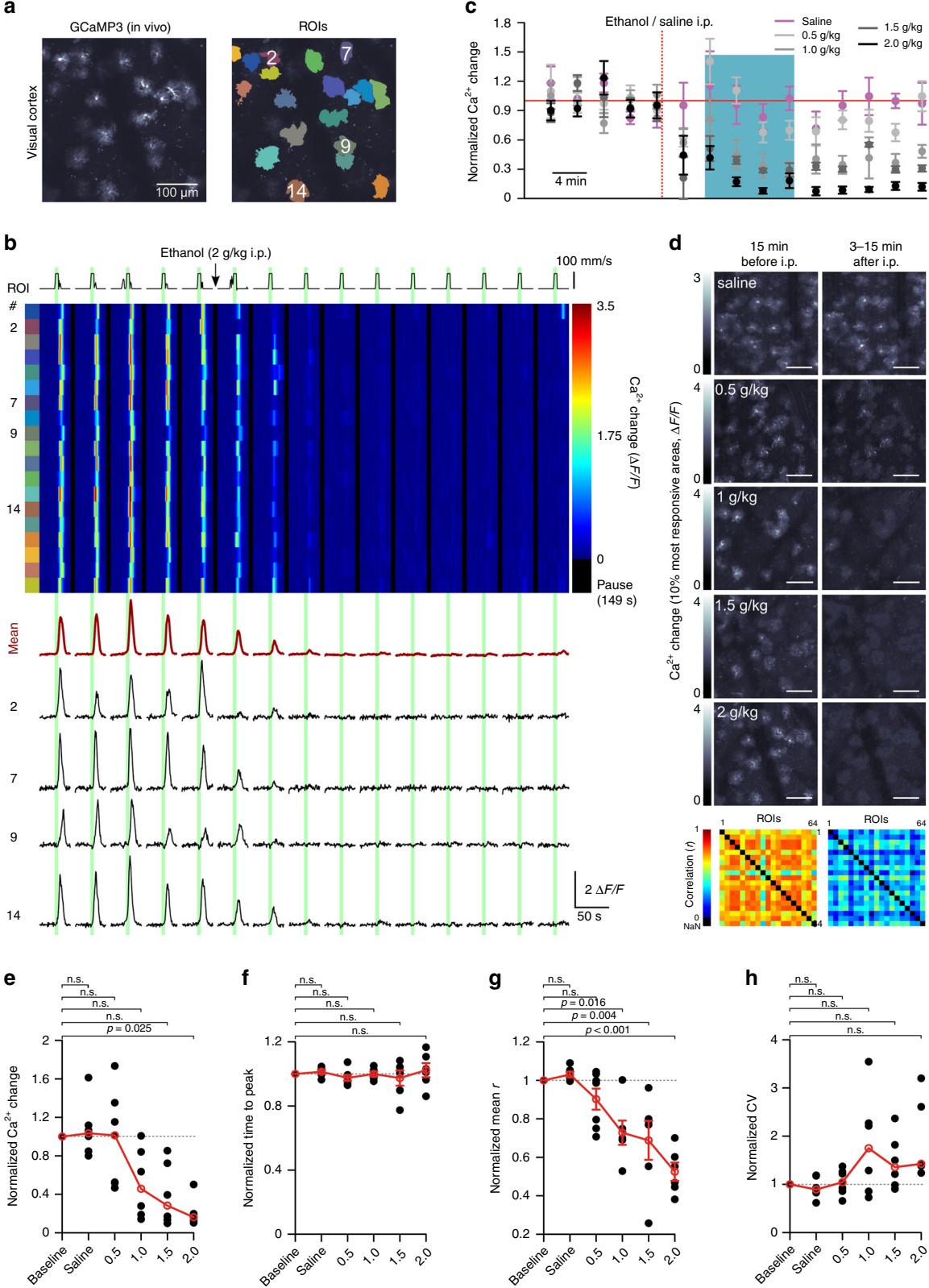

(Fig. 10b, c and Supplementary Movies 3 and 4), suggesting that BG activation is not required for accurate motor coordination. Moreover, restoring BG activation by co-injecting MK912 with ethanol did not improve motor coordination during ethanol exposure (Fig. 10d), and neither baseline motor coordination nor its sensitivity to ethanol were affected by ablating NE-dependent BG activation (Fig. 10e).

## Discussion

Neither the identity and location of the adrenergic receptor that is required for vigilance-dependent astroglia $Ca^{2+}$ activation nor the sensitivity of this receptor to any modulation of NE release is currently known. The considerable variability of voluntary locomotion-induced astroglia activation[7] suggests that adrenergic receptors are not saturated by NE released during natural

**Fig. 6 Ethanol impairs vigilance-dependent primary visual cortex astrocyte $Ca^{2+}$ activation. a** Left, In vivo $Ca^{2+}$ image of astrocytes in tangential optical section of primary visual cortex in *Slc1a3-CreER^T;R26-lsl-GCaMP3* mouse (31 independent experimental repetitions with similar results). Right, ROIs representing individual astrocytes. **b** Upper, Pseudocolour plot of ROIs' $Ca^{2+}$ dynamics. Green bars, enforced locomotion. Lower, Average $Ca^{2+}$ response of all ROIs (dark red) and $Ca^{2+}$ response traces within number-defined ROIs. **c** Time course of effect of saline/ethanol injection (red dotted line) on $\Delta F/F_{10s}$. Normalized to average first five trials (baseline). Blue bar, effect analysis time window. mean ± SEM; $n = 6$ mice (saline), 7 mice (0.5 g/kg), 6 mice (1.0 g/kg), 6 mice (1.5 g/kg), and 6 mice (2.0 g/kg). **d** Upper, Maximum response plots during indicated experimental episodes. Scale bar, 100 μm. Lower, Linear Pearson correlation coefficient plots between individual ROIs' $Ca^{2+}$ change traces before and after 2 g/kg ethanol. Numbers of independent experimental repetitions with similar results, respectively, were 6 (saline), 7 (0.5 g/kg), 6 (1.0 g/kg), 6 (1.5 g/kg), and 6 (2.0 g/kg). **e–h** Population data, mean normalized values within blue bar in **c**. Numbers under abscissa, g/kg i.p. ethanol. Data represent: mean $\Delta F/F_{10s}$ (**e**), mean time to peak (**f**), mean correlation coefficient ($r$) (**g**), and coefficient of variation (CV) (**h**) among $\Delta F/F$ traces of individual ROIs. Red symbols, median (**e**, **h**), or mean ± SEM (**f**, **g**). Lines between dots support readability. $n = 6$ mice (saline), 7 mice (0.5 g/kg), 6 mice (1.0 g/kg), 6 mice (1.5 g/kg), and 6 mice (2.0 g/kg); Kruskal–Wallis test (**e**, **h**) and one-way ANOVA (**f** $F(5, 32) = 0.512$, $p = 0.765$; **g** $F(5, 32) = 12.445$, $p < 0.001$) were followed by Tukey–Kramer correction; n.s. not significant. Source data are provided as a Source Data file.

behavior. Thus, astroglia activation might be affected even by moderate modulation of NE release. Despite numerous reports of interactions between ethanol and the LC–NE system, and rising interest in the role of astroglia in alcohol use disorders[52–56] the consequences of ethanol exposure for brain circuit NE targets, and in particular, the interaction between ethanol and vigilance-dependent astroglia function, have not been explored. By combining *Adra1a* null mutant with BG-specific *Adra1a* knockout experiments we found that BG activation requires $\alpha_{1A}$-adrenergic receptors (Figs. 3 and 4 and Supplementary Figs. 4 and 5). This study reveals that acute exposure to ethanol robustly inhibits locomotion-induced excitation of noradrenergic neurons (Fig. 9) resulting in reduced noradrenergic terminal $Ca^{2+}$ elevations (Fig. 8 and Supplementary Fig. 7) and reversible impairment of BG and V1 astrocyte activation (Figs. 1, 2 and 6), suggesting that ethanol inhibits phasic release of NE and brain-wide astroglia activation. We find that ethanol inhibition of noradrenergic neurons is sufficient to account for the impairment of BG activation, since pharmacological restoration of both signals following ethanol exposure could be achieved when NE release was restored through block of $\alpha_2$-adrenergic autoreceptors that normally inhibit NE release (Figs. 7, 8 and 9b and Supplementary Fig. 6) and topical application of ethanol to the cerebellar cortex did not inhibit BG activation (Fig. 9a). Finally, gait analysis revealed that the inhibition of BG $Ca^{2+}$ activation does not contribute to the impairing effect of ethanol on motor coordination (Fig. 10), suggesting that it may rather affect non-motor functions of the cerebellum[57]. In summary, our findings reveal how acute ethanol silences astroglia, a universal circuit element, during vigilance-dependent adjustments in neural activity.

While at least one of the three mammalian $\alpha_1$-adrenergic receptor subtypes is critical for astroglia activation[7,9], the receptor identity and location have not been known. In zebrafish, $\alpha_{1B}$-adrenergic receptors are required for radial glia to undergo $Ca^{2+}$ activation and suppress unsuccessful behavior[58]. In invertebrates, the analogous signaling molecule to NE is octopamine, and octopamine mediates arousal-induced direct $Ca^{2+}$ activation via receptors on glia[59]. We found that global knockout of $\alpha_{1A}$-adrenergic receptors was sufficient to impair BG activation (Fig. 3). BG- or astroglia-specific conditional knockout of $\alpha_{1A}$-adrenergic receptors, despite almost complete loss of *Adra1a* mRNA in the cerebellum (Supplementary Fig. 5), reduced those responses considerably, but incompletely (Fig. 4a, b and Supplementary Fig. 4a). This could be due to extended stability of the $\alpha_{1A}$-adrenergic receptor on the cell surface. It is also possible that truncation of the seventh transmembrane domain, the intracellular C-terminus, and 3 out of 17 key amino acid residues for the putative ligand binding site in *Adra1a^{cKO/cKO}* mice could preserve partially functional receptors[46]. Compensatory upregulation of expression of another $\alpha_1$-adrenergic receptor on BG cannot be

excluded, but it is not very likely considering the almost complete loss of BG responsiveness to NE in global $\alpha_{1A}$-adrenergic receptor knockout mice. An alternative interpretation of variable loss of NE responsiveness among BG would be variations in BG expression of this receptor[60]. However, the high degree of homogeneity in BG $Ca^{2+}$ responsiveness to locomotion regarding size and kinetics reflected by correlation between ROIs (Fig. 1) and the uniform loss of responsiveness in global $\alpha_{1A}$-adrenergic receptor knockout mice (Fig. 3) suggest that BG $\alpha_{1A}$-adrenergic receptors mediate $Ca^{2+}$ activation in response to locomotion-triggered NE release.

The direct modulation of astroglia activity by NE release suggests that these cells are particularly sensitive to any alteration of activity in LC or in other noradrenergic nuclei projecting to the cerebellum or forebrain[3]. Stress-induced expression of the immediate early gene c-Fos is attenuated in the hippocampus upon ethanol exposure[61], but noradrenergic neurons within LC increase c-Fos expression after acute ethanol exposure[62]. However, a diversity of stimuli and consecutive intracellular signaling pathways control c-Fos expression, and poor correlation between spiking activity and c-Fos expression has been found for spontaneously active neurons[63]. The most direct measure of NE release is through microdialysis where the limitations are spatial and temporal resolution, and the invasiveness of placing the sample collection device close to the brain area of interest[64]. Fast-scan cyclic voltammetry is an electrochemical approach to monitor neurotransmitter dynamics at subsecond time resolution[65]; however, despite considerable recent progress it can be challenging to distinguish between the detection of dopamine and NE[66]. Both, microdialysis and fast-scan cyclic voltammetry experiments revealed that depending on the brain region and experimental paradigm acute ethanol can lower, raise or leave extracellular NE concentration unchanged[26–28,37].

Our approach here has been to shed light on the acute effect of ethanol on noradrenergic signaling by imaging $Ca^{2+}$ dynamics on both sides of locomotion-induced direct signaling from noradrenergic terminals to astroglia. For monitoring axonal terminal $Ca^{2+}$ dynamics in awake behaving mice we reasoned that $Ca^{2+}$ dynamics in thin axonal terminals that might to a considerable proportion originate from voltage-gated $Ca^{2+}$ channels would be more readily detectable using a membrane-anchored GECI. Indeed, Lck-GCaMP6f overexpressed in noradrenergic terminals revealed robust locomotion-induced $Ca^{2+}$ elevations (Fig. 8 and Supplementary Fig. 7). Several lines of evidence support our model that ethanol inhibits noradrenergic neuron excitability and locomotion-induced terminal $Ca^{2+}$ elevations sufficiently to reduce NE release to the extent that $\alpha_{1A}$-adrenergic receptors on astroglia fail to trigger $Ca^{2+}$ elevations. (1) If ethanol increased extracellular NE through increased release or reduced reuptake, $\alpha_{1A}$-adrenergic receptors on astroglia could become

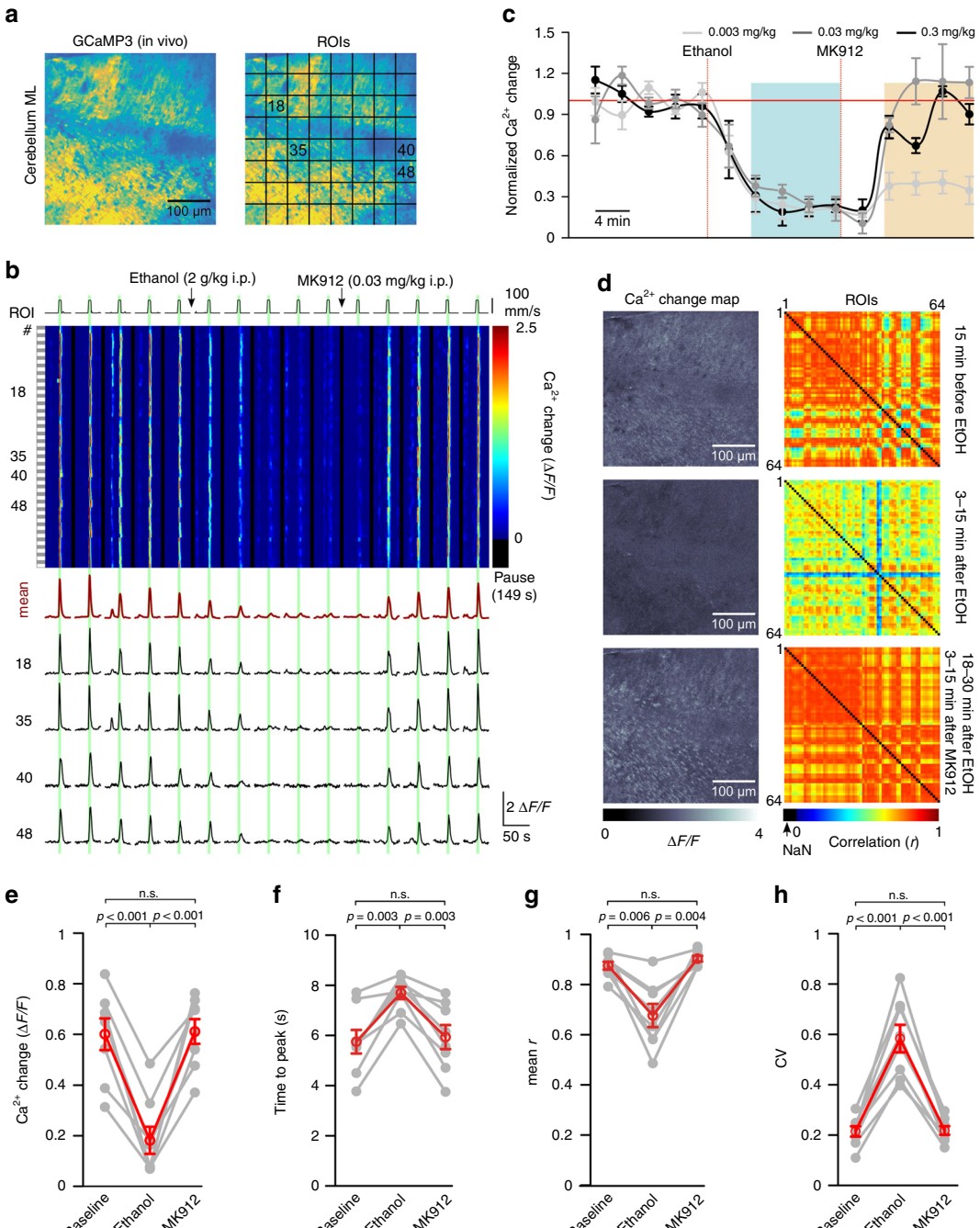

**Fig. 7 Pharmacological facilitation of NE release restores ethanol-inhibited vigilance-dependent cerebellar BG Ca²⁺ activation. a** Left, In vivo Ca²⁺ image of BG processes in *Slc1a3-CreER^T;R26-lsl-GCaMP3* mouse. Right, Locations of ROIs. Twenty independent experimental repetitions with similar results. **b** Upper, Pseudocolour plot of ROIs' Ca²⁺ dynamics. Green bars, enforced locomotion. Lower, Average Ca²⁺ response of all ROIs (dark red) and Ca²⁺ response traces within number-defined ROIs. **c** Time course of effect of ethanol (2 g/kg i.p.) injection followed by MK912 injection at indicated dosages (red dotted lines) on $\Delta F/F_{10s}$. Normalized to average first five trials (baseline). Blue bar, ethanol effect analysis time window; light brown bar, MK912 effect analysis time window. mean ± SEM; $n = 6$ mice (0.003 mg/kg), 8 mice (0.03 mg/kg) and 6 mice (0.3 mg/kg). Lines between dots support readability. **d** Left, Maximum response plots during indicated experimental episodes. Right, Respective pairwise linear Pearson correlation coefficient plots between individual ROIs' Ca²⁺ change traces during indicated experimental episodes (MK912, 0.03 mg/kg; eight independent experimental repetitions with similar results). **e–h** Population data, mean values within ethanol and 0.03 mg/kg MK912 time windows in **c**. Data represent: mean $\Delta F/F_{10s}$ (**e**), mean time to peak (**f**), mean correlation coefficient (r) (**g**) and coefficient of variation (CV) among $\Delta F/F$ traces of individual ROIs (**h**). Red symbols, mean ± SEM. Lines between dots support readability. $n = 8$ mice; repeated measures ANOVA (**e** $F(2, 14) = 49.137$; **f** $F(2, 14) = 23.347$; **g** $F(2, 14) = 22.108$; and **h** $F(2, 14) = 65.828$) was followed by Tukey–Kramer correction; n.s. not significant. Source data are provided as a Source Data file.

desensitized[67]; however, α₁ₐ-adrenergic receptor desensitization would not be consistent with our MK912 recovery experiment, since facilitation of noradrenergic terminal excitability and NE release would further support receptor desensitization (Fig. 7). (2)

Ethanol inhibited locomotion-induced noradrenergic terminal Ca²⁺ elevations, which is consistent with a model of reduced NE release (Fig. 8). (3) If due to the promiscuous pharmacological profile of ethanol[44], reduction of NE release was only one of

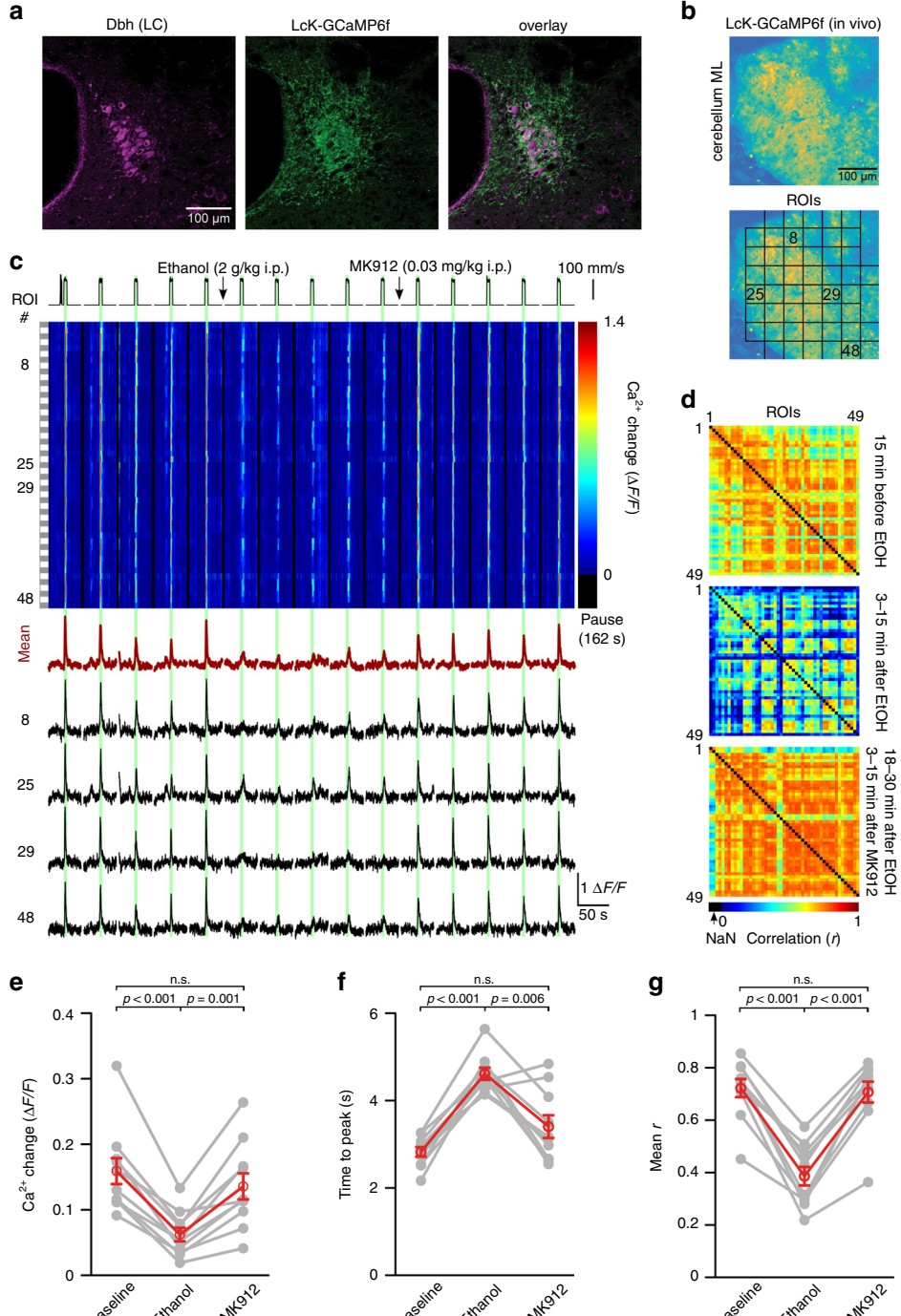

**Fig. 8 Ethanol inhibition of vigilance-dependent Ca$^{2+}$ elevations in NE terminals accounts for loss of BG Ca$^{2+}$ responsiveness. a** Confocal image of 6-week-old *Dbh-Cre;Lck-GCaMP6f$^{flox}$* mouse LC stained for eGFP (Lck-GCaMP6f, green) and dopamine β-hydroxylase (Dbh) (magenta). Three independent experimental repetitions were obtained with similar results. **b** Upper, In vivo image of *Dbh-Cre;Lck-GCaMP6f$^{flox}$* mouse cerebellar molecular layer. Lower, Locations of ROIs. Nineteen independent experimental repetitions were obtained with similar results. **c** Upper, Pseudocolour plot of ROIs' Ca$^{2+}$ dynamics. Green bars, enforced locomotion. Lower, Average Ca$^{2+}$ response of all ROIs (dark red) and Ca$^{2+}$ response traces within number-defined ROIs.
**d** Respective pairwise linear Pearson correlation coefficient plots between individual ROIs' Ca$^{2+}$ change traces during indicated experimental episodes.
**e–g** Population data representing mean $\Delta F/F_{10s}$ (**e**), mean time to peak (**f**), and mean correlation coefficient ($r$) (**g**) among $\Delta F/F$ traces of individual ROIs. Red symbols, mean ± SEM. Lines between dots support readability. $n = 10$ mice; repeated measures ANOVA (**e** $F_{(2, 18)} = 29.575$; **f** $F_{(2, 18)} = 27.698$; **g** $F_{(2, 18)} = 49.194$) was followed by Tukey–Kramer correction; n.s. not significant. Source data are provided as a Source Data file.

multiple converging mechanisms that resulted in loss of astroglia activation we would predict that MK912 would need to facilitate NE release in the presence of ethanol beyond baseline levels to restore normal astroglia activation. However, we found that the same dosage of MK912 that was sufficient to completely restore

BG activation, restored NE terminal Ca$^{2+}$ elevations just to the baseline level (Fig. 8), indicating that ethanol-induced reduction in NE release is sufficient to account for loss of astroglia activation. (4) Our findings that topical application of ethanol to the cerebellar cortex did not inhibit BG activation, whereas topical

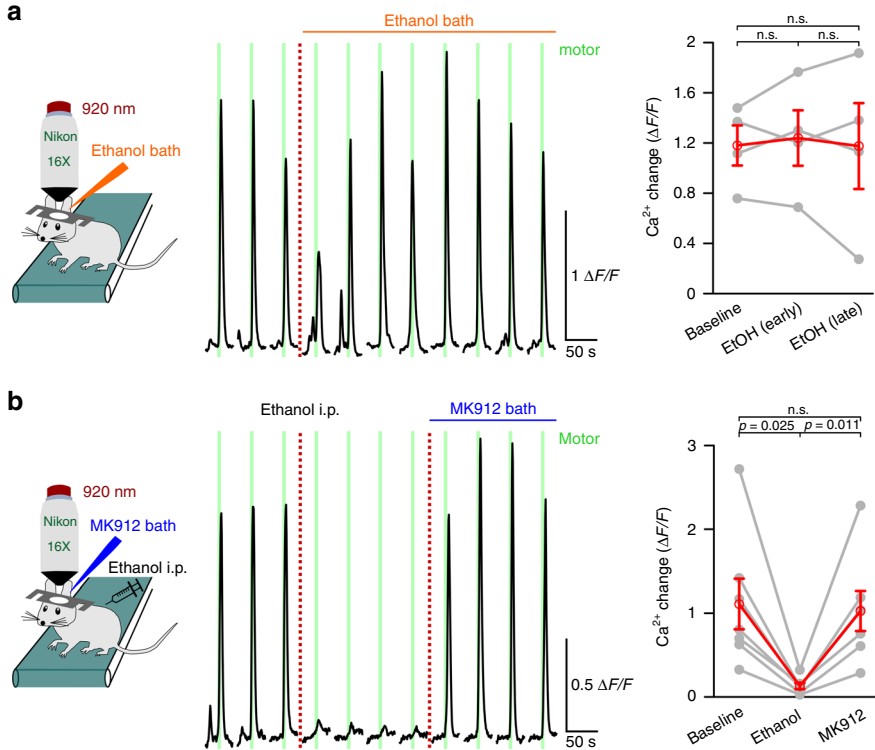

**Fig. 9 Ethanol inhibits the noradrenergic system at noradrenergic neuron somata or dendritic input but not at the terminals. a** Left, scheme of the awake topical pharmacology experimental design. Middle, representative average $Ca^{2+}$ response trace of BG processes of *Aldh1l1-CreER^{T2};Ai95* mouse. Ethanol bath concentration was 30 mM. Green bars, enforced locomotion. Right, Population data, mean $\Delta F/F_{10s}$ ± SEM of trials 1–3, 5–7, and 9–11, respectively. $n = 4$ mice; repeated measures ANOVA ($F(2, 6) = 0.114$) was followed by Tukey–Kramer correction. **b** as **a**; however, ethanol was injected (2 g/kg i.p.) and MK912 bath concentration was 15 μM. Population data, mean $\Delta F/F_{10s}$ ± SEM of trials 1–3, 5–7, and 9–11, respectively. $n = 7$ mice; repeated measures ANOVA ($F(2, 12) = 14.419$) was followed by Tukey–Kramer correction; n.s. not significant. Source data are provided as a Source Data file.

application of MK912 was sufficient to restore BG activation following inhibition by i.p. ethanol (Fig. 9), suggest that ethanol inhibits noradrenergic neurons at the soma or dendritic input rather than at the terminals, and they further suggest that inhibition of BG activation cannot be accounted for by any known target of ethanol in the cerebellar cortex[50]. Our gait analysis provided several lines of evidence that vigilance-dependent BG $Ca^{2+}$ activation is not required for motor coordination. (1) Disruption of alpha$_{1A}$ adrenergic signaling did not impair motor coordination. (2) Preservation of BG activation following acute ethanol exposure by MK912 did not prevent ethanol-induced ataxia. (3) 45 min after acute ethanol exposure, when BG activation was still disrupted, the mice performed already normally on the CatWalk XT. Together these findings suggest that ethanol inhibition of vigilance-dependent astroglia $Ca^{2+}$ activation could contribute to the impairing effect of ethanol on non-motor performance.

Our findings that acute ethanol exposure abolishes vigilance-dependent astroglia $Ca^{2+}$ activation through inhibition of NE release and subsequent failure of astroglia α$_{1A}$-adrenergic receptor activation provide mechanistic insight that help disentangle the complex effects of ethanol on attention deficit, motor discoordination, and addiction. Intriguingly, opioids, another group of highly addictive recreational drugs, have been shown to inhibit LC neuron excitability[68,69]. Therefore, the results of this study may serve as a blueprint to predict modulation of vigilance-dependent astroglia $Ca^{2+}$ activation and direct NE to neuron signaling by other recreational drugs. While our demonstration of limited completeness of gene deletion (Fig. 4 and Supplementary Fig. 4) needs to be considered when evaluating possible downstream consequences of astroglia $Ca^{2+}$ activation, α$_{1A}$-adrenergic

receptors in astroglia engage the same signaling pathways that are recruited by exogenous astroglia activation via opto- or chemogenetic approaches that have demonstrated the potential for profound behavioral and circuit consequences[12,14,43]. The sensitivity of vigilance-dependent astroglia $Ca^{2+}$ activation to recreational drugs will guide further investigations into the functional role of this ubiquitous CNS circuit motif during natural behaviors.

## Methods

**Experimental design.** All animal procedures were conducted in accordance with guidelines and protocols of the University of Texas Health Science Center at San Antonio (UTHSCSA) Institutional Animal Care and Use Committee. At the time of data collection all mice were between 2 and 6 months old. The ethanol sensitivity appeared stable within this age range (Supplementary Fig. 8a). For all awake behavior experiments we employed a previously reported head-restrained mouse on motorized treadmill paradigm[7]. Locomotion protocols as described in the respective figures and subsequent data analysis routines described below were automated, minimizing the risk of experimenter bias. In global or conditional knockout experiments the experimenter was left unaware of the mouse genotype. Acute cerebellar slice experiments were prepared from cerebella harvested from mice that had participated in awake behavior experiments at least 2 weeks earlier, with slices being taken from the hemisphere opposite to the chronic cranial window. On the basis of availability, male and female mice were assigned randomly to individual experiments and all datasets contain data from mice of both sexes. Sex-disaggregated data analysis did not reveal any trend that could be accounted for by a sex effect (Supplementary Fig. 8). For most experiments (Figs. 1–3, 6–10 and Supplementary Figs. 1, 2 and 5–9) the statistical analysis was based on number of mice. For experiments with more than one FOV per mouse analyzed (Figs. 4 and 5 and Supplementary Fig. 4 in vivo data) the statistical analysis was based on number of FOVs from at least three mice. For acute cerebellar slice experiments (Figs. 4 and 5 and Supplementary Fig. 4) statistical analysis was based on single-cell analysis. The mouse and equipment icons used in Figs. 1a and 9a, b, and the air puff and mouse silhouette used in Supplementary Fig. 1a were created by L.Y.

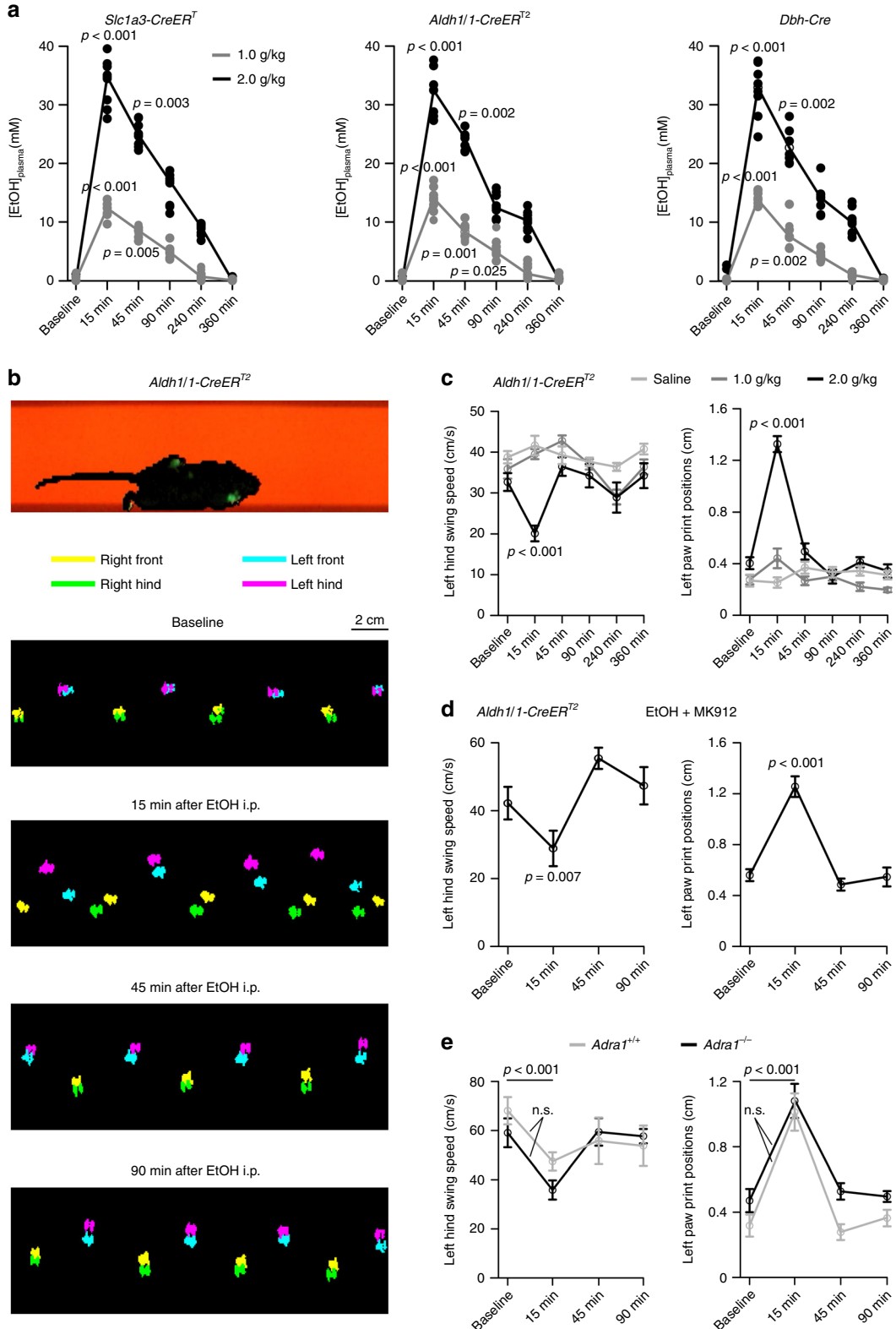

**Animals**. Mice were kept in the Laboratory Animal Resources facility with the ambient temperature maintained at 72–78 °F, the humidity at 30–70%, and the mice had ad libitum access to water and chow. Mice were maintained on a reverse 12h-light/12h-dark schedule (lights off at 9 a.m., on at 9 p.m.) and all experimental procedures were completed during the dark cycle. The transgenic mouse breeding strategy followed the following rules: For all experiments either GCaMP3, GCaMP6f (Ai95 (ref. [70])) or Lck-GCaMP6f was expressed as GECI in a Cre-dependent manner. For the parameters analyzed in this study, the slight differences in kinetics between astroglia $Ca^{2+}$ dynamics measured with GCaMP3 or GCaMP6f were not relevant[71]. Each animal was heterozygous for one GECI allele and heterozygous for one Cre recombinase allele. For global as well as conditional knockout experiments the final breeding step was arranged so that offspring heterozygous for GECI, heterozygous for Cre recombinase, and either homozygous for the mutant or floxed allele or homozygous for the wt allele was possible. *Adra1a* mutant or floxed homozygous mice appeared healthy and were born in a frequency expected from Mendelian principles. All genotypes were determined from extracted toe or tail sample DNA via polymerase chain reaction (PCR) using the primers listed in

**Fig. 10 Loss of vigilance-dependent BG Ca$^{2+}$ activation does not contribute to ethanol-induced ataxic motor coordination. a** Time course of plasma ethanol concentration following i.p. injection of 1 or 2 g/kg in *Slc1a3-CreER$^T$;Ai95*, *Aldh1l1-CreER$^{T2}$;Ai95*, and *Dbh-Cre;Lck-GCaMP6f$^{flox}$* mice ($n = 8$, respectively). For statistical analysis, Friedman test was used followed by Tukey–Kramer correction. **b** Upper, representative CatWalk XT raw data where any contact area of the mouse with the track appears green. Lower, paw prints with color-coded paw identity under baseline conditions or at indicated amounts of time following 2 g/kg i.p. ethanol. **c** Quantification of paw swing speed in *Aldh1l1-CreER$^{T2}$* mice, dependent on time and dosage of ethanol exposure (left, time: $F_{(5, 110)} = 4.834$; ethanol dosage: $F_{(10,110)} = 3.911$) and precision of paw positioning (right, time: $F_{(5, 110)} = 28.871$; ethanol dosage: $F_{(10, 110)} = 22.108$), mean ± SEM from 8 mice (saline), 8 mice (1.0 g/kg), and 9 mice (2.0 g/kg), three between-subjects groups repeated measures ANOVA followed by Tukey–Kramer correction. **d** as **c**; however, the effect of 2 g/kg i.p. ethanol in combination with 0.03 mg/kg i.p. MK912 on paw swing speed ($F_{(3, 18)} = 10.226$) and on precision of paw positioning ($F_{(3, 18)} = 31.480$) was tested, mean ± SEM from 7 mice, repeated measures ANOVA followed by Tukey–Kramer correction. **e** as **c**; however, the effect of deletion of $\alpha_{1A}$-adrenergic receptor on sensitivity to 2 g/kg i.p. ethanol was tested, mean ± SEM from 7 mice (*Adra1a$^{+/+}$*) and from 6 mice (*Adra1a$^{-/-}$*); mixed between (genotype) and within (baseline versus 15 min) subjects ANOVA. n.s. not significant. Source data are provided as a Source Data file.

Supplementary Table 1. The *Adra1a$^{cKO}$* mouse line generated for this study will be made available upon request.

**Generation and genotyping of the *Adra1a$^{cKO}$* mouse line**. Adrenergic receptor $\alpha_{1a}$ (*Adra1a*) conditional mouse mutants were generated based on "knockout-first allele" strategy with a promoter-driven LacZ reporter. Depending on the use of site-specific recombinase such as Cre and Flp, this strategy allowed us to create null and conditional alleles of *Adra1a*, and knockout allele expressing LacZ under control of endogenous Adra1a promoter (see Supplementary Fig. 3). To generate the conditional allele of *Adra1a*, a part of intron 2, the entire exon 3, and a part of intron 3 were flanked by loxP sites. Two embryonic stem (ES) cell (JM8A3.N1) clones derived from C57bl/6N-A/a mice, where Adra1a allele was targeted by homologous recombination using "knockout-first allele" strategy were purchased from the European conditional mouse mutagenesis (EUCOMM) program. To confirm the proper targeting of the *Adra1a* allele, we characterized both ES cell clones by a PCR-based strategy (Supplementary Fig. 3b). We karyotyped these clones to confirm that they have normal chromosomal content. One correctly targeted ES cell clone (# HEPD0601_6_G07; project ID: 46217) was used to generate chimeric mice by injecting these cells into a C57bl/6N-derived blastocyst. The germline transmission was confirmed by breeding these mice to albino C57bl/6N mice that were purchased from the Charles River Laboratories. The heterozygote founders for *Adra1a* "knockout-first allele" were bred to Flp deleter mice to generate *Adra1a* conditional mutants (*Adra1a$^{cKO}$*). *Adra1a$^{cKO}$* mutants were further crossed 2–3 generations to C57bl/6N mice to establish this mouse line. Additionally, we designed a genotyping PCR-based strategy to be able to differentiate between wild type (383 bp), knockout-first (242 bp), conditional (496 and 242 bp), and null (501 bp) alleles of the *Adra1a* gene using a single PCR reaction (Supplementary Fig. 3). The primer details are listed in Supplementary Table 1.

**Tamoxifen administration and recombination efficiency**. Tamoxifen (Sigma-Aldrich, #T5648-5G) was freshly dissolved in sunflower seed oil (Sigma-Aldrich, #1642347-1G) at a concentration of 10 mg/ml by vortexing and sonication for approximately 15 min. It was usually stored in the refrigerator for up to 5 days; for conditional knockout experiments it was stored for up to 2 days. For experiments using the *Slc1a3-CreER$^T$* mouse line to overexpress GCaMP3 or 6f (Figs. 1, 3, 6, 7 and 10 and Supplementary Figs. 6 and 8) tamoxifen was injected intraperitoneally (10 μl per g mouse body weight starting at age 3–4 weeks) three times within 5 days, using a 1 ml syringe and a 25G 5/8-inch needle (BD PrecisionGlide Needle). This resulted in GCaMP3 or 6f overexpression in all BG and approximately 35% of primary visual cortex astrocytes. For experiments using the *Aldh1l1-CreER$^{T2}$* mouse line to overexpress GCaMP6f (Figs. 2, 5, 9, 10 and Supplementary Figs. 1 and 2) tamoxifen was injected intraperitoneally (10 μl per g mouse body weight starting at age 3–4 weeks) three times within 5 days, using a 1 ml syringe and a 25G 5/8-inch needle (BD PrecisionGlide Needle). This resulted in GCaMP6f overexpression in all BG. For experiments using the *Dbh-Cre* mouse line to overexpress Lck-GCaMP6f (Figs. 8, 10 and Supplementary Figs. 7 and 8) in noradrenergic neurons, no tamoxifen administration was necessary and from immunofluorescence analysis of LC we estimated a recombination efficiency close to 100% (Fig. 8a). For experiments using the Jdhu/J *Pcp2-Cre* mouse line (Jackson stock # 010536) to overexpress GCaMP6f in cerebellar Purkinje cells (Supplementary Fig. 5), no tamoxifen was necessary to obtain expression in all Purkinje cells. For experiments using the *Tg(ma6-cre)B1Lfr* mouse line to overexpress Lck-GCaMP6f in cerebellar granule cells, no tamoxifen was necessary to obtain expression in all granule cells. Surgeries were performed 1–2 weeks after the last tamoxifen injection and experiments started at least 2 weeks following surgeries. To maximize the probability of Cre recombination in *Adra1a$^{cKO/cKO}$* mice (Fig. 4 and Supplementary Figs. 4 and 5), tamoxifen administration was extended to five injections within 9 days. On average, this resulted in 8.2% NE-unresponsive BG in acute slices using the *Gli1-CreER$^{T2}$* mouse line, and 9.7% NE-unresponsive BG in acute slices using the *Aldh1l1-CreER$^{T2}$* mouse line. With conditional knockout experiments, imaging was conducted at earliest 1 month (3–4.5 months for slice experiments, 6–7 weeks

for RT-qPCR) following the last tamoxifen injection to allow for RNA and protein degradation following gene deletion.

**Animal surgery**. Surgery procedures had two steps. In the first-step surgery, under intraperitoneal anesthesia (100 mg/kg ketamine and 10 mg/kg xylazine in 0.9% saline solution), the mouse was placed on a heating pad to keep body temperature at ~36 °C. Following removal of the hair, the skin was disinfected with povidone-iodine. The skin and muscles were removed from the skull, 3% hydrogen peroxide was applied to further aid in disinfection and prevent bleeding. The periosteum was shaved off and approximately 3 mm of muscle surrounding the exposed skull was covered with a thin layer of cyanoacrylate cement. A custom-designed stainless steel head-plate with a 4 mm × 6 mm oval opening was centered on the skull above lobulus simplex/crus I of the cerebellar hemisphere (Figs. 1–4, 7–9 and Supplementary Figs. 2, 4 and 6–8), or above vermis at the cerebellar midline (Supplementary Fig. 1), or above primary visual cortex at lambda, 2.5 mm lateral from midline (Fig. 6), and was mounted on the skull using dental cement (C&B Metabond, Parkell Inc., Brentwood). While mice recovered from anesthesia, we coated wound edges with Neosporin® ointment and placed the mouse in a heated cage for recovery. In the second-step surgery, under isoflurane anesthesia (1.5–2% vol./vol. isoflurane in O$_2$ with flow rate adjusted according to hindpaw pinch reflex) and on a heating pad, a craniotomy was performed on a skull area of 2.5 mm × 2.5 mm, the dura mater was removed and replaced with three fused layers of No. 1 cover glass. The edges of glass were gently sealed by dental cement (Ortho-Jet-Acrylic-Powder, Lang). After 1 week of postsurgery recovery, mice were habituated to the linear treadmill and the recording conditions in at least 3 sessions, each matching the later experiment with two exceptions: Saline was injected i.p. instead of drugs and the laser was off. Imaging was performed at least 2 weeks after surgeries. The topical pharmacology experiments were conducted at least 5 days after the first-step surgery, and after two habituation sessions.

**Immunohistochemistry**. Under intraperitoneal anesthesia (100 mg/kg ketamine and 10 mg/kg xylazine in 0.9% saline solution), mice were perfused by cardiac puncture with ice-cold 4% paraformaldehyde (Polysciences, Inc.) in 0.1 M phosphate buffer saline (PBS). Brains were removed from the skull and immersed in the same fixative solution for 4 h, at 4 °C. The brain tissue was kept in PBS containing 0.1% sodium azide at 4 °C. Parasagittal cerebellum sections were cut on a vibratome (VF-300-OZ, Precisionary Instruments). Cerebellum sections (thickness of 35 μm) were soaked in the 0.1 M PBS for 10 min and incubated for 3 h in blocking solution with 0.1 M PBS, 5% normal goat serum (NGS) (Jackson ImmunoResearch Laboratories Inc.), and 1% Triton X-100 (Sigma-Aldrich). Next, the sections were incubated with the primary antibodies diluted in the blocking solution containing 5% NGS in 0.1 M PBS with 0.5% Triton X-100 for 36 h at 4 °C. The primary antibodies were used at following concentrations: chicken anti-eGFP (1:1000, # A10262, polyclonal, Thermo Fisher Scientific) (Figs. 4a and 8a) and mouse (IgG$_1$) anti-S100β (1:500, # MA1-25005, monoclonal (SH-B4), Thermo Fisher Scientific) (Fig. 4a) or rabbit anti-DBH (1:500, # 22806, polyclonal, ImmunoStar) (Fig. 8a). The sections were washed four times with 0.1 M PBS containing 5% NGS for 10 min, respectively, and incubated for 4 h at room temperature in 0.1 M PBS with 5% NGS with appropriate fluorescence-conjugated secondary antibodies: Alexa Fluor® 488-conjugated AffiniPure goat anti-Chicken IgY (1:5000, #103-545-155, Jackson ImmunoResearch) (Figs. 4a and 8a) and Alexa Fluor® 647-conjugated AffiniPure Goat Anti-Mouse IgG$_1$ (1:5000, #115-605-205, Jackson ImmunoResearch) (Fig. 4a) or Alexa Fluor® 647 AffiniPure Goat Anti-Rabbit IgG (H + L) (1:5000, #111-605-144, Jackson ImmunoResearch) (Fig. 8a). The sections were then rinsed once with 0.1 M PBS containing 5% NGS for 10 min and three times with 0.1 M PBS for 10 min, respectively. The slices were mounted on microscope glass slides using Aqua-Poly/Mount coverslipping medium (#18606-20, Polysciences, Inc.) and dried overnight. The next morning, they were sealed with two-component epoxy adhesive (BSI-201, Bob Smith Industries, Inc.).

**Confocal imaging**. The confocal imaging data were obtained using a Zeiss LSM 710 confocal microscope. For cerebellum whole parasagittal sections, tiled scanning was used with a ×10 objective (Plan-Apochromat, 0.45 NA, Zeiss); 5 Z-stack slices with 4 μm step size at <1 airy unit pinhole setting (Fig. 4a). For higher magnification images a ×40 oil immersion objective (EC Plan-Neofluar, 1.3 NA, Zeiss) was used; 12 Z-stack slices with 4 μm step size at <1 airy unit pinhole setting (Figs. 4a and 8a). Images represent maximum intensity projections of image stacks. We used the 488 nm line of an Argon laser to excite Alexa Fluor® 488 and the 633 nm line to excite Alexa Fluor® 647.

**Slice preparation**. Two- to 5-month-old mice were anesthetized by isoflurane inhalation and decapitated. The cerebellum was mounted on the flat specimen holder of a vibratome (Leica VT1200, Leica Biosystems) by tissue adhesive seal (#1469SB, 3M Vetbond) leaning against 2% agarose in saline. The ice-cold cutting solution was saturated with 95% $O_2$ and 5% $CO_2$ and contained (in mM, pH 7.4): 135 $C_7H_{17}NO_5$ NMDG (N-methyl-D-glucamine), 1 KCl, 1.2 $KH_2PO_4$, 1.5 $MgCl_2$, 0.5 $CaCl_2$, 20 $C_5H_{14}NO·HCO_3$ (choline bicarbonate), 11 $C_6H_{12}O_6$ (dextrose). Parasagittal cerebellum slices were cut at 250 μm thickness and collected at 37 °C in oxygenated (95% $O_2$, 5% $CO_2$) artificial cerebrospinal fluid (aCSF) containing (in mM, pH 7.4): 119 NaCl, 2.5 KCl, 1 $NaH_2PO_4$, 1.3 $MgCl_2$, 2 $CaCl_2$, 26.2 $NaHCO_3$, 11 $C_6H_{12}O_6$ (dextrose). The beaker with collected slices was transferred to room temperature after completion of the slicing while the solution continued to be constantly bubbled with 95% $O_2$ and 5% $CO_2$. Before being transferred to a recording chamber, the respective slice was incubated in aCSF (constantly bubbled with 95% $O_2$ and 5% $CO_2$) containing 1 μM TTX (Tocris Bioscience, #1069) for ~30 min.

**In vivo 2P imaging**. Two setups were used for $Ca^{2+}$ imaging experiments. For all chronic astroglia in vivo experiments (Figs. 1–4, 6, 7 and Supplementary Figs. 1, 2, 4, 6 and 8), we used a galvanometer-based 2P laser-scanning Movable Objective Microscope (MOM) (Sutter Instruments) with a ×16, 0.80 NA water-immersion objective (Nikon). A pulsed Ti:Sapphire laser beam at 920 nm, 80 MHz repetition rate, 140 fs pulse width (Coherent Inc., Chameleon Ultra II) was focused at approximately 60 μm below the pial surface reaching half-way through the cerebellar molecular layer or deep into layer 1 of primary visual cortex. To avoid excessive brain injury the laser power was attenuated to 12–35 mW at the front aperture of the objective. Emitted light was detected using a photomultiplier tube (H10770PA-40; Hamamatsu Photonics). The awake mouse was placed on the custom-made linear treadmill (Fig. 1a)[7] and the head-plate was mounted under the microscope objective. The speed of the treadmill belt could be monitored with an optical encoder. The belt was freely movable so that the mouse could walk voluntarily; however, at predefined episodes a servo motor could be engaged to enforce locomotion at 80–110 mm/s. In Supplementary Fig. 1, at alternating predefined episodes, the animal was subjected to enforced locomotion or a 40 psi air puff to its whiskers. Electromyography (EMG) signals were recorded as the surface potential difference between two silver wires inserted subcutaneously at the right shoulder and left hip of the mouse. The 2P microscope was controlled by an Xi Computer Corporation personal computer (Intel(R) Core(TM) i7-4930 CPU @ 3.40 GHz, 8 GB of RAM) running ScanImage (v3.8.1; Vidrio Technologies, LLC) software within MATLAB R2011b (Mathworks). Image acquisition was triggered at a rate of 1.5 frames/s, and locomotion speed data, EMG data, and Y-mirror position data were simultaneously acquired at 20 kHz sampling rate using National Instruments boards controlled by custom-written scripts in Labview2013 (version 13.0.1f2, National Instruments). Acquired frames were 400 μm × 400 μm at a resolution of 512 pixels per line and 512 lines per frame. Non-imaging data were post hoc downsampled to the image acquisition frame rate and the Y-mirror signal was used to assign appropriate data bins to individual image frames. The entire experimental setup was enclosed in a blackout box. For experiments on noradrenergic terminal $Ca^{2+}$ dynamics and for topical pharmacology experiments (Figs. 8, 9 and Supplementary Fig. 7), a resonant scanning version of the MOM (Sutter Instruments) was used with a ×16, 0.80 NA water-immersion objective (Nikon). A pulsed Ti:Sapphire laser beam at 920 nm, 80 MHz repetition rate, <120 fs pulse width (Insight DS+, Spectra-Physics MKS Instruments Light & Motion) was used for 2P excitation. The acquisition rate was 15 frames/s with the laser power adjusted to 9–16 mW at the front aperture of the objective. The microscope was controlled by an Xi Computer Corporation personal computer (Intel(R) Core(TM) i7-5930K CPU @ 3.50 GHz, 16 GB of RAM) running ScanImage (v5.0; Vidrio Technologies, LLC) software within MATLAB R2016a (Mathworks). All other design principles including the treadmill were matched between microscopes.

**2P imaging in acute brain slices**. For acute slice 2P $Ca^{2+}$ imaging experiments (Figs. 4 and 5 and Supplementary Fig. 4) we used the same microscope and imaging settings as described above for in vivo $Ca^{2+}$ imaging of vigilance-dependent astroglia $Ca^{2+}$ dynamics, except that we replaced the treadmill with a slice recording chamber. The laser beam tuned to 920 nm was focused to visualize as many BG somata as possible within the 400 μm × 400 μm (512 pixels × 512 pixels) FOV. Laser power was set to 10–15 mW at the front objective aperture. The slices in the recording chamber were continuously superfused with aCSF at room

temperature containing 1 μM TTX. DL-Norepinephrine hydrochloride (NE, 30 μM) (#A7256-1G, Sigma-Aldrich) and adenosine 5′-triphosphate disodium salt hydrate (ATP, 100 μM) (#2383-1G, Sigma-Aldrich) were bath-applied sequentially on the same slice. NE was applied during 2 min after baseline recordings (2 min) in aCSF solution. Following 15 min of NE washout, ATP was applied for 2 min. Both NE and ATP testing solutions contained 1 μM TTX. We intentionally used the same agonist application order (first NE, then ATP) based on our observation that despite the 15 min washout of the first agonist of a $G_q$ protein-coupled receptor, the application of a second agonist of a different $G_q$ protein-coupled receptor usually resulted in a weaker $Ca^{2+}$ elevation (Fig. 4c and Supplementary Fig. 4b). Therefore, we reasoned that a stronger response to the second agonist would be a reliable indication of a functionally relevant knock-down of the receptor of the first agonist.

**Pharmacology**. To maintain a minimally invasive experimental approach, all drugs were applied systemically via the intraperitoneal (i.p.) route. The following drug preparations were used and administered at 100 μl/10 g body weight: For a target dosage of 2 g/kg ethanol, 127 μl ethanol was mixed with 373 μl saline, and for 1.5, 1.0, and 0.5 g/kg the mixture was adjusted proportionally. For a target dosage of 0.03 mg/kg MK912, 3 mg of MK912 was dissolved in 1 ml of saline and then diluted 1000× in saline. For target dosages of 0.3 and 0.003 mg/kg of MK912, the final dilution was adjusted proportionally. For control i.p. injections, 100 μl saline/ 10 g body weight was injected. Injections were conducted immediately following the previous imaging trial. The same FOV was imaged before and after i.p. injections. For prazosin hydrochloride (Cat. No. 0623, Tocris), 1 mg of prazosin was dissolved in 100 μl DMSO (276855, Sigma-Aldrich) + 100 μl Tween 80 (P4780, Sigma-Aldrich) and 800 μl saline to achieve the final dosage of 10 mg/kg. Injections were conducted immediately following the previous imaging trial. Acute superficial application (Fig. 9): For topical bath application of the drug, ethanol or MK912 was dissolved in aCSF at a final concentration of 30 mM and 15 μM, respectively. In order to optimize access of drugs to the brain while preserving integrity of the tissue following removal of the skull we removed the dura and continuously superfused the surface of the cerebellum with aCSF heated to 36–37 °C. In order to dampen tissue motion associated with the mouse walking on the treadmill we cemented a 3 mm (rostrocaudal direction) × 800 μm strip of No. 1 cover glass across the skull window. An additional 2 mm × 800 μm of No. 1 cover glass was fused in the center of the larger glass described above using UV curable optical adhesive. This procedure provided sufficient gentle pressure against the surface of the cerebellum to enable 2P imaging during locomotion events while providing access for diffusion of drugs into the imaged tissue. The same FOV was imaged before and after i.p. injection or bath application of drugs. Animals were head-fixed for no more than 90 min during a single imaging session.

**Catwalk motor performance analysis**. Motor performance of the mice was assessed using the Catwalk XT system (Noldus, Wageningen, Netherlands). *Aldh1l1-CreER^T2;Ai95* mice aged P90–120 and *Slc1a3-CreER^T;Ai95;Adra1a^{+/+}* or *Adra1a^{−/−}* mice aged P60–90 (Fig. 10) with equal sex distribution were placed on an enclosed glass walkway (20 × 10 cm) illuminated from top by a red fluorescent light and along the walkway by green light-emitting diodes. Paw prints were tracked and measured automatically by interruption of the green light from the bottom, while disruption of red light allowed for visualization of images of the mice. Mice were allowed to walk freely across the walkway, with bedding and treats in both ends encouraging the animals to walk continuously. All animals were habituated to the walkway at least 10 min for 2 days before experiments. For Fig. 10b, c, *Aldh1l1-CreER^T2;Ai95* mice were divided into three groups: saline ($n =$ 8 mice), 1 g/kg ethanol i.p. injection ($n = 8$ mice), and 2 g/kg ethanol i.p. injection ($n = 9$ mice). For each group, data were acquired for baseline (before i.p.) and 15, 45, 90, 240, and 360 min after i.p. injection. For Fig. 10d *Aldh1l1-CreER^T2;Ai95* mice were administered a cocktail of 2 g/kg ethanol and 0.03 mg/kg MK912 i.p. after baseline, then at 15, 45, and 90 min motor performance was recorded. For Fig. 10e *Aldh1l1-CreER^T2;Ai95;Adra1a^{+/+}* or *Adra1a^{−/−}* animals were administered 2 g/kg ethanol i.p. after baseline, then at 15, 45, and 90 motor performance was recorded. A digital high-speed camera recorded each run until three compliant runs had been acquired in which the animal crossed the walkway within 0.5–10 s with a maximum speed variation of less than 60%. After each trial, the walkway was wiped down with double-distilled water. Catwalk XT 8.1 software (Noldus) was used to analyze the data. For each animal, parameters were analyzed independently for left and right hind paws, including swing speed (mean speed of the paw during swing) and print position (mean distance between the centroid of front paw print and respective centroid of hindpaw print (normally close to 0)), parameters that have been reported to be affected by ethanol[51].

**Plasma ethanol concentration measurement**. Animals from the following lines, including *Slc1a3-CreER^T;Ai95, Aldh1l1-CreER^T2;Ai95,* and *Dbh-Cre;Lck-GCaMP6f^{flox}*, aged from P90 to 150 with equal sex distribution were anesthetized via intraperitoneal anesthesia described above and placed on a heating pad to maintain body temperature. Blood samples were collected from retro-orbital sinus at baseline (before injection) and 15, 45, 90, 240, and 360 min after ethanol injection. Samples (50–100 μl venous blood) were collected in Eppendorf tubes that had been wetted with 5 μl of 5000 IU/ml heparin in water. Plasma was obtained by

centrifugation (Eppendorf Centrifuge 5415D) of whole blood at $9246 \times g$ for 2 min, then kept at 4 °C, and measured within 24 h to minimize evaporation of ethanol. The ethanol concentration was determined using an Ethanol Assay Kit (K620-100, Biovision) following the manufacturer's instructions, using the colourimetric method with a microplate reader (Synergy HT, BioTek). For each sample, the plasma was serially diluted to 1:250, 1:1000, and 1:2500, including the reaction buffer. The final plasma concentration was calculated as mean of the concentration values obtained from all dilutions that yielded readings within the linear range of the ethanol calibration curve.

**RT-qPCR.** Conditional knockout mice ($Aldh1l1$-$CreER^{T2}$;$Adra1a^{cKO/cKO}$) and wild-type littermates ($Aldh1l1$-$CreER^{T2}$;$Adra1a^{wt/wt}$) or global knockout mice ($Adra1a^{-/-}$) and wild-type littermates ($Adra1a^{+/+}$) at 3 months old ($n = 3$ in all groups), were used for RT-qPCR after tamoxifen treatment (described above). Mice were sacrificed by decapitation and cerebellum was excised and as much brainstem removed as possible. Half of the tissue was snap-frozen in liquid nitrogen and stored at −80 °C for subsequent western blot analysis. Total RNA was extracted using GeneJET RNA Purification kit (Thermo Fisher K0731) and DNA removed using amplification grade DNase I (Invitrogen 18068-015). Reverse-transcribed cDNA was generated from 500 ng RNA so that the concentration of cDNA in the final qPCR reaction was (assuming 1:1 ratio of RNA to cDNA) 37.5 ng per sample. Reverse transcription was done using SuperScript III First-Strand Synthesis System for RT-PCR (Invitrogen 18080-051) with a 1:1 mixture of oligo-dT primers and random hexamers. The cDNA was amplified for $Adra1a$, $Slc1a3$, and $Atp5b$. For qPCR, triplicate samples were prepared with SYBR Green Master Mix (Thermo Fisher A25742) and primers for $Adra1a$, $Slc1a3$, and $Atp5b$. The primer details are listed in Supplementary Table 1. The following program was run in a 7900HT Fast Real-Time PCR System: 50 °C for 2:00 (minutes:seconds), 95 °C for 10:00, [start cycle] 95 °C for 0:10, 60 °C for 0:25, 72 °C for 0:35 [end cycle; repeat 40 cycles], 95 °C for 0:15, 60 °C for 1:00, 95 °C for 0:30, and 60 °C for 0:15.

### Data analysis

*Locomotion-induced $Ca^{2+}$ dynamics.* Imaging data were saved in ScanImage as tiff files, imported to MATLAB R2011b or R2016a, and stored as mat files. Analysis was conducted using custom-written scripts employing a combination of built-in, open-source, and custom-written functions. Any computer code will be shared upon request. Images were first passed through a Gaussian filter (1.52 SD per pixel distance) to attenuate random noise of the detector. Individual frames within the entire time series (including all imaging trials with each trial representing: baseline imaging frames + imaging frames during 5 s locomotion episode + imaging frames until the next imaging pause highlighted by black bars in all pseudocoloured plots) were registered to maximize correlation. For cerebellar BG processes or noradrenergic terminals, where individual structure elements could not be assigned to individual cells, ROIs were spatially defined by an $8 \times 8$ checkerboard pattern, which resulted in ROIs of approximately 50 μm × 50 μm (Figs. 1–3, 7–9 and Supplementary Figs. 1, 2 and 6–8). For V1 experiments (Fig. 6) ROIs were defined by individual astrocytes. For the analysis of in vivo conditional knockout experiments, we chose to select the entire FOV as sole ROI since otherwise the analysis would have been biased towards responsive BGs, independent of their proportional representation within the total BG population.

Data from imaging noradrenergic terminals were averaged (every two consecutive image frames) for noise reduction, resulting in a final effective frame rate of 7.5 Hz. For each trial and ROI we performed the following analysis: We subtracted the average detector offset obtained from imaging under identical conditions with the laser shutter closed. We determined the median fluorescence value during baseline (from start of a trial until the frame before onset of locomotion), then calculated the $\Delta F/F$ fluorescence values for each image frame by $(F - F_{median})/F_{median}$, with $F$ being the absolute mean fluorescence value within a given image frame. We previously found that voluntary locomotion-induced BG $Ca^{2+}$ elevations during the baseline episode can suppress enforced locomotion-induced responses[7]. Therefore, we determined the average standard deviation (SD) of baseline $\Delta F/F$ values of all trials of a data set. We then used the criterion, if at least three consecutive $\Delta F/F$ values within a baseline exceeded 3 × SD, this trial was considered contaminated by voluntary locomotion and it was excluded from quantification. The term "mean $Ca^{2+}$ change" represents mean $\Delta F/F$ within 10 s from onset of locomotion of the average of all ROIs' $\Delta F/F$ traces. "Mean Time to peak" (TTP) represents the time from onset of locomotion to the maximum $\Delta F/F$ value within 10 s from onset of locomotion of the average of all ROIs' $\Delta F/F$ traces. The "mean correlation coefficient" represents the mean Pearson's linear correlation coefficient among all pairs of ROI $\Delta F/F$ traces within 20 s from onset of locomotion. The "coefficient of variation" represents the standard deviation among all ROIs' mean $\Delta F/F$ values within 10 s from onset of locomotion, divided by the mean of all ROIs' mean $\Delta F/F$ values within 10 s from onset of locomotion. In $Adra1a^{-/-}$ mice residual locomotion-induced BG $Ca^{2+}$ elevations were considerably slower than in wild-type littermates; therefore, we adapted our analysis strategy by first testing if a response was detectable (using the same strategy described above for detecting trials contaminated by voluntary activity-induced $Ca^{2+}$ elevations during baseline) from onset of locomotion until the end of the trial. If a response was detectable we determined the "mean time to peak". For "mean $Ca^{2+}$ change" and "coefficient of variation" we conducted above

described analysis from 5 s before until 5 s after time of peak response. For "mean correlation coefficient" we conducted above described analysis from 10 s before until 10 s after time of peak response. For each experiment, values from trials that were not contaminated by voluntary activity of the mouse were averaged. Following any drug administration, we discarded the first trial following i.p. injection to allow the drug to reach its effective concentration in the brain and averaged all remaining trials that were not contaminated by voluntary mouse activity. In cases where data are presented normalized, they were normalized to the average of baseline trials' values.

*Agonist-induced $Ca^{2+}$ dynamics in acute cerebellar slices.* Slice imaging data were saved in ScanImage v3.8.1 as tiff files, imported to MATLAB R2011b or R2016a, and stored as mat files. For visualization of relative responsiveness of BG to NE or ATP we calculated a representative $\Delta F/F$ image frame via the following steps: To obtain a representative baseline fluorescence image frame, for each of the $512 \times 512$ pixels of the raw image time series we calculated the mean of the 20 brightest respective pixel values within the 177 baseline image frames. To obtain a representative response fluorescence image frame, for each of the $512 \times 512$ pixels of the raw image time series we calculated the mean of the 20 brightest values of the respective pixel within the 277 image frames from onset of bath application of the respective agonist (covering the 2 min agonist application episode and reaching approximately 68 s into washout of the agonist, since the timing of the peak response is variable in slice bath application experiments). For each agonist the $\Delta F/F$ image was calculated as (response$_{i,j}$−baseline$_{i,j}$)/baseline$_{i,j}$, with $i$ and $j$ representing the pixel indices. The final visualization of relative responsiveness of BG to NE or ATP, as presented in Fig. 4c and Supplementary Fig. 4b, was then obtained by subtracting the ATP $\Delta F/F$ image from the NE $\Delta F/F$ image. BG that responded stronger to NE than to the consecutively applied ATP appear brighter and have pixel values >0 in this visualization. This is expected as typical signal for a wild-type BG response, since both NE and ATP cause BG $Ca^{2+}$ elevations in a $G_q$ protein-coupled receptor-dependent manner and incomplete recovery of $G_q$ protein-coupled receptor downstream signaling steps usually diminishes consecutive responses. If responses were equally strong to both agonists, or if no responses occurred at all (see background areas above the pial surface of the molecular layer and the granule cell layer that is devoid of GCaMP6f expression in the $Gli1$-$CreER^{T2}$ mouse; Fig. 4a, c) pixel values were around 0. A negative pixel value on the other hand indicates that a cell responded less to NE compared to ATP even though NE was at the advantage of a fully recovered $G_q$ protein-coupled receptor downstream signaling pathway, suggesting that noradrenergic receptors were knocked down to a functionally relevant extent. In order to analyze individual BG $Ca^{2+}$ responses we defined ROIs representing individual BG responsive to ATP in the ATP $\Delta F/F$ image. Since some BG may have had a weak response to ATP due to a previous strong response to NE, we then superimposed the identified ROIs on the corresponding NE $\Delta F/F$ image and added ROIs of cells that appeared responsive but were not yet accounted for by the ROIs. We used the criterion described above to detect trials contaminated by voluntary locomotion to detect BG unresponsive to ATP following a response to NE (were eliminated from analysis) or unresponsive to NE despite being responsive to consecutive application of ATP. In order to calculate the ratio of responsiveness to NE over ATP of an individual ROI, we applied the following strategy that is analogous to our analysis of vigilance-dependent astroglia $Ca^{2+}$ dynamics: We calculated the $\Delta F/F$ trace based on the median fluorescence value during baseline (177 frames−2 min). BG $Ca^{2+}$ responses to bath application of NE or ATP were slow, usually lasting more than one minute, and either monophasic or oscillatory in shape; thus, to obtain a representative mean $\Delta F/F$ value for each ROI and agonist, the 50% highest fluorescence values during the 2 min bath application as well as the initial 2 min of the 15 min washout episode were averaged. mean $\Delta F/F_{NE}$/mean $\Delta F/F_{ATP}$ ratio values >1 indicate BG responded stronger to NE, mean $\Delta F/F_{NE}$/mean $\Delta F/F_{ATP}$ ratio values <1 indicate BG responded stronger to ATP even though it was applied after NE exposure. Analysis of slice experiments in Fig. 5 was conducted accordingly, however with reversed agonist order.

*Analysis of RT-qPCR.* Relative mRNA expression level of $Adra1a$ and control $Slc1a3$ between $Adra1a^{cKO/cKO}$ or $Adra1a^{-/-}$ and $Adra1a^{wt/wt}$ mice was quantified by $\Delta\Delta$Ct analysis: For each triplicate ($Adra1a^{cKO/cKO}$) or quadruplicate ($Adra1a^{-/-}$) of qPCR reactions for $Adra1a$, $Slc1a3$, and $Atp5b$ (housekeeping reference) per mouse the mean cycle threshold (Ct) value was determined. For each mouse, the $\Delta$Ct value for $Adra1a$ and for $Slc1a3$ was determined by subtracting the mean Ct value of $Atp5b$ from the mean Ct value of $Adra1a$ or $Slc1a3$, respectively. The mean percentage of $Adra1a$ or $Slc1a3$ mRNA present in $Adra1a^{cKO/cKO}$ or $Adra1a^{-/-}$ mice compared to $Adra1a^{wt/wt}$ mice was calculated as $2^{-(\text{mean}\Delta Ct_{X},Adra1ac^{KO/cKO} \text{ or } Adra1a^{-/-} - \text{mean}\Delta Ct_{X},Adra1a^{wt/wt})} \times 100\%$, with $X$ representing $Adra1a$ or $Slc1a3$.

*Statistical analysis.* Statistical analyses were performed using MATLAB R2016a (MathWorks). For each group of a data set the Lilliefors test was applied to test for Gaussian distribution. If all groups followed a Gaussian distribution we reported and presented the data as mean ± SEM. If two groups were compared, we applied the two-tailed Student's $t$-test; for more than two unrelated groups, we applied the one-way ANOVA, and if groups were related, we applied the repeated measures

ANOVA. If at least one group did not follow Gaussian distribution we reported and presented the data as median and range. If two or more unrelated groups were compared, we applied the Kruskal–Wallis test; if more than two groups were compared that were related, we applied the Friedman test. For all tests that involved more than one comparison, we applied the Tukey–Kramer correction for multiple comparisons. For the mixed between- and within-subjects ANOVA in Fig. 10e we used https://www.mathworks.com/matlabcentral/fileexchange/27080-mixed-between-within-subjects-anova. The basis of the sample number for individual tests and respective test type applied are mentioned in the results section, usually the respective figure legend. The significance of test results is indicated by $p$ values in the graphs or by the abbreviation n.s. for not significant when $p \geq 0.05$.

**Reporting summary**. Further information on research design is available in the Nature Research Reporting Summary linked to this article.

## Data availability

All datasets generated and/or analyzed during the current study are provided as a Source Data file. Source data are provided with this paper.

## Code availability

The MATLAB codes generated for data analysis and display during the current study are available at https://github.com/PaukertLab/NCOMMS-20-01106/find/main, and from the corresponding author on reasonable request. Source data are provided with this paper.

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

## Acknowledgements

The authors would like to thank Priscilla M. Barba-Escobedo, John Cavaretta, and Naiqing Ye for expert support in genotyping and animal husbandry, and Drs. Louis Reichardt and Jason Pugh for sharing the *Tg(ma6-cre)B1LFR* mouse line. A.A. is supported by the Chica and Heinz Schaller Research Foundation and grant A09N/SFB1158 from the Deutsche Forschungsgemeinschaft. This work was supported by R01MH083728 to D.E.B., and R01AA025128, R01MH113780, and The Robert J. Kleberg, Jr. and Helen C. Kleberg Foundation to M.P.

## Author contributions

L.Y. designed, performed and analyzed most experiments and contributed to manuscript writing. M.O. performed and analyzed most experiments and contributed to manuscript writing. X.Z. performed and analyzed all NE terminal experiments. E.Y.L. performed and analyzed the recombination efficiency evaluation of the *Adra1a^cKO* mouse line. R.R.D. optimized primers for the RT-qPCR. A.A. generated and genomically analyzed the *Adra1a^cKO* mouse line, and designed the RT-qPCR strategy. D.E.B. supervised the generation of the *Adra1a^cKO* mouse line. M.A.B. provided expert advice on immunocytochemistry experiments. M.P. conceived the project, supervised all experiments, aided in data analysis and wrote the manuscript. All authors discussed the results and commented on the manuscript.

## Competing interests

The authors declare no competing interests.
