## [Peer Review File · Nature Communications]

Reviewers' comments:

Reviewer #1 (Remarks to the Author):

This is an interesting paper that examines the mechanism by which ethanol blocks locomotion-induced astrocyte calcium signaling. The paper demonstrates that ethanol abolishes locomotion induced calcium increases in cerebellar Bergman Glia (BG) and V1 astrocytes, that loss of α 1A-adrenergic receptors globally and specifically in BG abolish locomotion-induced astrocyte signaling, that ethanol does not interfere with responsiveness of BG but blocking α 2-adrenergic receptors restores the responsiveness of BG and finally that ethanol reduces the activity of NE terminals in the cerebellum. The authors use a nice complement of approaches including multiphoton slice and in vivo calcium imaging and pharmacology, mouse genetics and chemogenetic approaches to determine that loss of the vigilance-dependent calcium elevations in astrocytes is mediated by reduced activity and release of NE from LC projecting neurons into the cerebellum and V1.

As noted by the authors, the role of NE in astrocyte vigilance-dependent calcium signaling has been previously shown as was the effect of ethanol on the activity of noradrenergic LC neurons and NE levels. Therefore, although some nice experiments are described, this study offers no significant conceptual advance.

Some suggestions for improving the manuscript are listed below:

1. Although the authors very briefly mention the role of astrocytes in alcohol use disorders (AUD) in the discussion, it would improve the manuscript to include some introduction on what is known about the impact of acute or long-term ethanol exposure on astrocytes and its relevance to AUD.
2. Blood ethanol concentration (BEC) measurements are strikingly absent in this study, considering it is an alcohol paper. Could the variability in the data with different doses on some of the analyzed parameters (Fig 1f-i) be attributed to different BEC achieved in different animals? It will therefore be important to show the BECs attained at the different injected doses. Also, it is important to verify that the transgenic manipulations (Gli1-CreERT2;Ai95, ALDH1L1-CreERT2;Ai95 and Dbh-Cre;Lck-GCaMP6fflox mice) does not alter ethanol pharmacokinetics.
3. Authors mention that n=30 (14 mice) for data shown in Figure 1f-i. Is this from combining data from the different ethanol doses (1.5g/kg and 2.0g/kg)?
4. There are some inconsistencies in the way data for the same parameters are reported throughout the results section- sometimes mean \pm SEM sometimes range.
4. Can the authors describe the total number of habituation sessions and the duration of each session? In Figure 2, they perform several imaging sessions on the same day. How long did each imaging session last?
5. The methods section mentions that all the studies were performed between 2 to 6 months of age. With such a huge age range for experiments, could this contribute to variability in the extent of recombination? Have they considered the possibility that the late adolescent and adult mice display differences in their sensitivity to acute alcohol exposure.
6. Male and female mice were used in the study and authors mention that all data sets contain data from mice of both genders. The authors report that gender-disaggregated data analysis did not reveal any trend that could account for gender effect. Were sufficient number of mice used for the analysis of sex differences? Could some of the variability that they observe in the Ca²⁺ parameters be due to differences in sensitivity to alcohol doses between males and females?

Therefore, a combination of different age ranges and genders could contribute to different sensitivities to alcohol.

7. The recombination efficiency is not reported for any part of this study.

8. Authors mention that a considerable fraction of BG responded less to NE than to subsequent ATP application in *Adra1a*CKO/cKO mice and that many BG exhibit no response to NE. Based on the traces shown in 4C (lower right), it appears that only 2 out of the 6 BG soma did not show a response to NE. Could they report the proportion of BG that responded to NE in both the WT and KO slices?

10. Not clear why the dura mater was removed following craniotomy as no drugs are applied locally.

Reviewer #2 (Remarks to the Author):

This manuscript presents experiments designed to improve our understanding of how alcohol (EtOH) affects locus coeruleus driven, adrenergic modulation of Bergmann glial cell astrocytes in the cerebellar cortex. The authors use multiphoton confocal fluorescence imaging of gCaMP (a genetically encoded Ca²⁺ sensitive fluorophore), in awake, but head fixed mice on a treadmill, to assay EtOH modulation of locomotion associated bergman glial (BG) Ca²⁺ signaling.

The authors report that forced locomotion (the treadmill moves, and given the head fix scenario, the mice are likely to ambulate, although consideration of EtOH-induced motor impairment is not adequately described or addressed) causes a robust increase in Ca²⁺ in cerebellar BG cells.

The authors provide a very thorough description of the variability of this response across micro domain regions, the kinetics and amplitude of such responses.

The authors then report that EtOH dose dependently (starting at 1mg/kg, and plateauing at 1.5-2 mg/kg, which likely equates to BECs of 21 to 42 mM; Porcu et al., 2010 ACER) blocks such Ca²⁺ responses. The blockade begins to recover by 2 hrs. and is fully reversed by 6 hours post EtOH injection.

Subsequent use of global and BG specific knockout of α_1 adrenergic receptors for further in vivo studies, and pharmacological studies in brain slices revealed that such in vivo effects are mediated by BG α_1 receptors, but that the modulation is not via direct EtOH actions on the α_1 receptors or downstream signaling. Instead, use of α_2 receptor antagonists to presumably increase locus coeruleus neuronal excitability, and parallel studies imaging Ca²⁺ in locus coeruleus axonal terminals within the cerebellar cortex generated data that was most compatible with EtOH block of BG Ca²⁺ signaling being mediated by suppression of NE release. I.e. increasing LC excitability with α_2 antagonists rescued EtOH block, and EtOH also blocked locomotion induced Ca²⁺ signaling in LC axonal terminals in parallel and with similar pharmacological sensitivity to BG responses.

Parallel experiments demonstrated that qualitatively similar processes occur in cortical astrocytes.

From these observations, the authors conclude that EtOH increases NE release from LC axonal terminals onto BGs which raises BG intracellular Ca²⁺, and likely also in any other astrocytes in the vicinity of LC axonal terminals throughout the brain. Given increasingly appreciated expanding roles of astrocytes in brain function, such effects are likely contributing mediators of overall EtOH actions on

the brain.

Overall, these are complicated and very nicely designed experiments, and the interpretation and presentation of the raw data is clear and compatible with the authors' conclusions. The issues explored are clinically important (alcoholism is a serious and prevalent societal issue), and the mechanisms explored will also be of broader interest to the general neuroscience community. The overall design, execution, analysis and presentation of experiments is sound.

Despite many positive aspects to this study, I have a number of considerations and concerns that lessen my enthusiasm with respect to publication in Nature Communications.

First and foremost, while this is an elegant study using sophisticated methodology, the main conclusion is that EtOH increases locus coeruleus output, which excites Bergmann Glia. This is of course important information, but it is also one of many actions of EtOH in the brain, but in this manuscript there is no exploration of cellular mechanism or role in any relevant alcohol related trait/behavior that adds to our understanding of alcohol abuse and addiction. Thus, while this study adds to our already large list of cellular actions of EtOH, it does not substantially change our understanding of how such actions occur or how these specific actions relate to any aspects of alcohol related behaviors or clinical issues. That is of course fine, but it doesn't substantially advance the broad field. There are tens if not hundreds of reports detailing EtOH increasing or decreasing excitability of various neuronal or glial populations, and many of them have been explored enough to determine whether such actions are mediated pre or post synaptically, or both, and what specific mechanism mediate such responses, and how those responses affect behavioral actions of EtOH.

Beyond that general issue, there are a number of conceptual and technical/experimental issues that complicate this story (please see specific comments below).

Specific comments.

1) Although my thoughts about writing style would normally be reserved for the minor comments section, in this case, I found the abstract and introduction, and related coverage of experimentation to be very confusing, generally, but especially in relation to what exactly is being studied. More specifically, in the abstract and introduction, the authors discuss/introduce astrocytic Ca²⁺ signaling in a very chaotic fashion, making it hard to understand what the authors think they are studying. The authors refer to/describe such signals and evoking behaviors as arousal, vigilance, sensory processing, locomotion-induced, adaptive regulation, attention, reward, motivation and stress, vigilant attention and cognitive performance. The authors then also describe how such glial signaling may be involved in development of AUD, brain development, adult function, synaptogenesis, pruning, ionic homeostasis, neurotransmitter transport, modulation of synaptic function, metabolic support and neurovascular coupling.

While many of these processes may be functionally related, overall this reads like a laundry list of astrocyte function without any guidance to how this leads to or relates to the ensuing elegant study.

While astrocytes, and their Ca²⁺ signaling are likely or clearly involved in these various functions (and many may be functionally linked), it is not clear how any are related to or drive the current studies of EtOH modulation of such signaling?

In parallel, there is no guidance or presentation of what is known about EtOH actions on astrocytes, and how the authors' experimentation will address any gaps. The authors can surely convey in 1-3 sentences that astrocytes do a lot more than we used to think they did, but should spend the rest of their writing leading us to the specifics of why they've focused on locus coeruleus, locomotion and their respective or interactive relationship to alcohol related behaviors.

2) In addition to the writing issues addressed above, it is actually important to consider experimentally what the recorded BG Ca²⁺ signals and modulation reflect behaviorally.

Important issues that should be addressed experimentally or at least with a coverage of clear evidence in the existing literature include:

A) Are such signals passive sensory evoked, motor output evoked, arousal evoked, vigilance evoked, or something else? Do such signals dictate behavior or are they responsive to behavior or both? And, whatever case can be made, the authors should be consistent in how they then refer to such signals.

B) Relatedly, why did the authors focus on "lobules simplex crus 1 of the hemispheres"? The case for how this subregion relates to what specific behavior the authors suggest they are studying should be made.

C) The authors should consider, explore and discuss how the above issues relate to the impact of EtOH generally, but in particular in the context of the blood concentration of alcohol that was likely achieved in the various studies. Alterations in BG signaling are first detected at 1mg/kg, and reach plateau at 1.5 and 2 mg/kg, which best estimation based on previous literature (Porcu et al. *ibid*) would be respectively 21, 30 and 42mM. What do such concentrations of EtOH do to rodent behavior? Certainly motor impairment, and at higher doses, loss of righting reflex. Does the associated change in BG Ca²⁺ signaling cause such motoric responses, or do alterations in such motoric responses cause the change in Ca²⁺ signaling? Given recent appreciation of cognitive/emotional roles of cerebellum (e.g. Carta et al. 2019 *Science*), are changes in BG signaling a reflection of reward, anxiolysis, aversion?

3) While the authors touch on the issue of other cells possibly contributing to the altered signals, there are serious complications to consider if the signals truly only come from LC NE release. First and foremost, such high doses of EtOH are known to affect GABA and glutamate release within the molecular layer, both of which are known to affect BG Ca²⁺ signaling. Why aren't such signals and alteration by EtOH playing a role here? Relatedly, norepinephrine is known to modulate Purkinje cell excitability and it likely interacts with EtOH, how are all of these factors interacting?

4) It would be informative to know whether actions on LC are mediated at the soma or on the terminals.

5) The behaviorally associated BG Ca²⁺ signals are surprisingly widespread and uniform. Are similar responses seen in other regions of the cerebellum, are they or can they be evoked by unrelated behaviors, and are such responses similarly modulated by EtOH or modulation of adrenergic receptors?

6) Given the clear ability to pharmacologically (or genetically: the BG specific α_1 KOs!) prevent EtOH suppression of BG Ca²⁺ signals, it would add enormous excitement to this manuscript, to determine if doing so affected some EtOH related behavior (especially consumption and/or motor impairment), which would powerfully link BG responses and the cerebellum to EtOH behavioral actions.

Reviewer #3 (Remarks to the Author):

The manuscript describes effects of ethanol on cerebellar and cortical astrocyte signaling that involve changes in norepinephrine release. The investigators used a variety of powerful approaches, including imaging of intracellular calcium changes in Bergmann glia, cortical astroglia and noradrenergic axons. using head-fixed, awake mice running on a treadmill. Slice experiments were also used to help characterize the ethanol effects on responsiveness to NE. Genetic and pharmacological manipulations

were used to provide evidence for the proposed role of changes in NE release in the effects of ethanol. In general, the findings are convincing and the scientific rigor is impressive. The ethanol dose-response curves are especially impressive, as is the full characterization of recovery following the ethanol action. However, potentially informative experiments were not performed, and there are questions about some of the approaches.

Major Comments:

1) The evidence for involvement of alpha1 adrenergic receptors in the locomotion-related BG calcium increases comes mainly from the experiments with knockout mice. However, as the authors indicate this knockout does not appear to be complete. Previous work from this group indicated involvement of NE and alpha1 receptors in these responses, but a nonspecific antagonist trazodone was used. Why not repeat these experiments with a selective alpha1 antagonist?

2) While it is clear that the ethanol treatment has a minimal and short-lasting effect on actual locomotor performance, it would be good to know how the other genetic and pharmacological manipulations affected movement on the treadmill. Ultimately, it is important to determine if glial calcium signaling is involved in locomotor function, or provides a general signal for attention or arousal. Obviously, this issue can't be resolved in the present study, but the author should discuss the subject.

3) The size of the ROIs shown in figure indicate that the authors weren't attempting to measure calcium changes at the single bouton level. Thus, it should be made clear to the reader that fields of axons and terminals were imaged in this experiment.

4) While the LC provides a strong NE projection to cerebellum, recent evidence suggests that other noradrenergic nuclei also project there (e.g. Robertson et al., Nature Neuroscience, 2013). Thus, the authors may want to be more inclusive in discussing which nuclei might contribute to the NE effects.

Minor Comments:

i. Figure 6C, it's not clear that the lines connecting the points serve any purpose. The color coding of the symbols is sufficient to convey the story. Also, the light gray symbols are hard to see against a white background, so another color should be used for the 0.5 g/kg group.

ii. It appears that the $\Delta F/F$ values were used to detect voluntary locomotion-induced calcium increases, which makes sense. However, was this corroborated by the EMG data that was also collected?

iii. Page 32, the authors mention the use of a Goat Anti-Mouse IgG1, but this reviewer did not see any mouse primary antibodies in the methods description.

iv. Page 38, line 855, the statement "...suggesting that noradrenergic receptors were knocked down to a functionally relevant extent." is not accurate, as the loss of signal is not necessarily due to loss of receptors, but may also include impaired signaling downstream of the receptor.

v. There are several grammar and wording issues that should be corrected, particularly in the methods section.

Reviewer #1:

This is an interesting paper that examines the mechanism by which ethanol blocks locomotion-induced astrocyte calcium signaling. The paper demonstrates that ethanol abolishes locomotion-induced calcium increases in cerebellar Bergman Glia (BG) and V1 astrocytes, that loss of $\alpha 1A$ -adrenergic receptors globally and specifically in BG abolish locomotion-induced astrocyte signaling, that ethanol does not interfere with responsiveness of BG but blocking $\alpha 2$ -adrenergic receptors restores the responsiveness of BG and finally that ethanol reduces the activity of NE terminals in the cerebellum. The authors use a nice complement of approaches including multiphoton slice and in vivo calcium imaging and pharmacology, mouse genetics and chemogenetic approaches to determine that loss of the vigilance-dependent calcium elevations in astrocytes is mediated by reduced activity and release of NE from LC projecting neurons into the cerebellum and V1.

As noted by the authors, the role of NE in astrocyte vigilance-dependent calcium signaling has been previously shown as was the effect of ethanol on the activity of noradrenergic LC neurons and NE levels. Therefore, although some nice experiments are described, this study offers no significant conceptual advance.

We have rewritten the Introduction to present the conceptual advances more clearly. We will further edit the Results and Discussion once all new data have been added to the manuscript.

Additional concerns:

1. Although the authors very briefly mention the role of astrocytes in alcohol use disorders (AUD) in the discussion, it would improve the manuscript to include some introduction on what is known about the impact of acute or long-term ethanol exposure on astrocytes and its relevance to AUD.

We have rewritten our Introduction to include additional information about what is known about acute ethanol toxicity on astrocytes. Unfortunately, there is a paucity of information, as all studies to date have been performed on cultured astrocytes, which are markedly different from astrocytes *in situ*.

2. Blood ethanol concentration (BEC) measurements are strikingly absent in this study, considering it is an alcohol paper. Could the variability in the data with different doses on some of the analyzed parameters (Fig 1f-i) be attributed to different BEC achieved in different animals? It will therefore be important to show the BECs attained at the different injected doses. Also, it is important to verify that the transgenic manipulations (Gli1-CreERT2;Ai95, ALDH1L1-CreERT2;Ai95 and Dbh-Cre;Lck-GCaMP6fflox mice) does not alter ethanol pharmacokinetics.

This is an excellent point. We used transgenic lines (ALDH1L1-CreERT2;Ai95 and Dbh-Cre;Lck-GCaMP6fflox) for ethanol experiments. Accordingly, we are now determining the BEC before and 15 min, 45 min, 90 min and 240 min after i.p. injection of saline or 1.0 g/kg or 2.0 g/kg ethanol in these lines.

3. Authors mention that n=30 (14 mice) for data shown in Figure 1f-i. Is this from combining data from the different ethanol doses (1.5g/kg and 2.0g/kg)?

We are now stating that the analysis of data shown in Figure 1f-i was based on "n = 6 per dosage of ethanol or saline". In analogy, we are now stating that the analysis of data shown in Figure 6e-h was based on "n = 6 - 7 per dosage of ethanol or saline".

4. There are some inconsistencies in the way data for the same parameters are reported throughout the results section- sometimes mean±SEM sometimes range.

We have added information to the Statistical Analysis paragraph in the Methods section as well as to the legend of Figure 1, indicating that mean±SEM was used to present and report data that followed a Gaussian distribution and were analyzed using parametric statistical tests, and that median and range was used to present and report data that did not follow a Gaussian distribution and were analyzed using non-parametric statistical tests.

5. Can the authors describe the total number of habituation sessions and the duration of each session? In Figure 2, they perform several imaging sessions on the same day. How long did each imaging session last?

These additional details have been added to the Methods. There were at least three habituation sessions during the second week following surgeries. Each habituation session matched the later experiment with two exceptions: Saline was injected i.p. instead of drugs and the laser remained off.

For the data shown in Figure 2, each session consisted of four to five locomotion trials at an interval of 200 s. This additional information has been added to the legend.

6. The methods section mentions that all the studies were performed between 2 to 6 months of age. With such a huge age range for experiments, could this contribute to variability in the extent of recombination? Have they considered the possibility that the late adolescent and adult mice display differences in their sensitivity to acute alcohol exposure.

While the overall age range for the study was indeed 2 to 6 months, the specific age range for the conditional α_{1A} -adrenergic receptor knockout slice experiments using the *Gli1-CreER^{T2}* Cre line was 160 - 180 days, and using the *ALDH1L1-CreER^{T2}* Cre line the age range was 120 - 150 days with very similar recombination efficiency (we have added the age ranges to the respective figure legends). We have also added a Supplementary Figure 6a that discloses the age of each mouse used for the ethanol sensitivity study in Figure 1 which indicates a very similar ethanol sensitivity within the age range investigated.

7. Male and female mice were used in the study and authors mention that all data sets contain data from mice of both genders. The authors report that gender-disaggregated data analysis did not reveal any trend that could account for gender effect. Were sufficient number of mice used for the analysis of sex differences? Could some of the variability that they observe in the Ca²⁺ parameters be due to differences in sensitivity to alcohol doses between males and females?

We have followed the NIH standard of analyzing all data gender-disaggregated to recognize any potential trend towards a gender-effect. Had such a trend been discovered we had increased the data size to test for a gender-effect. However, the absence of such a trend (we are illustrating representative examples of this analysis in new Supplementary Figure 6) did not justify further follow-up.

Therefore, a combination of different age ranges and genders could contribute to different

sensitivities to alcohol.

Our analysis in response to comments #6 and #7 suggests that age and gender do not underlie the variability in response of individuals. It needs to be emphasized that by presenting dot plots where each dot represents individual results, we have chosen the most transparent way of data presentation. Until recently, many studies have been reported using bar graphs which can lead to an underappreciation of the variability of the underlying data. Therefore, we are not convinced that our results, most of which represent awake mouse experiments, reflect exceptional variability.

8. The recombination efficiency is not reported for any part of this study.

We may consider two categories of recombination efficiency for this study. (1) Efficiency of overexpression of Rosa26 locus targeted, Cre recombinase-dependent genetically encoded calcium indicators (GECIs) - discussed here. And (2) efficiency of floxed alpha_{1A}-adrenergic receptor gene deletion - discussed in response to comment #9 below. For the overexpression of GECIs in cerebellar Bergmann glia, we employed Cre driver lines *Slc1a3-CreER^T*, *Gli1-CreER^{T2}* or *Aldh111-CreER^{T2}*. All of these lines produced ~100% recombination efficiency in Bergmann glia. For primary visual cortex astrocytes, we used the *Slc1a3-CreER^T* mouse line with a recombination efficiency that matched with ~35% our previous reports (PMID:24945771, PMID:28742117). For overexpression of Lck-GCaMP6f in noradrenergic neurons using the *Dbh-Cre* mouse line we estimate a recombination efficiency of ~100% based on our immunofluorescence analysis of locus coeruleus in Figure 8a. We have added this information to the "Tamoxifen administration" paragraph of the methods section, which we now term "Tamoxifen administration and recombination efficiency".

9. Authors mention that a considerable fraction of BG responded less to NE than to subsequent ATP application in *Adra1acKO/cKO* mice and that many BG exhibit no response to NE. Based on the traces shown in 4C (lower right), it appears that only 2 out of the 6 BG soma did not show a response to NE. Could they report the proportion of BG that responded to NE in both the WT and KO slices?

While Rosa26 targeted "floxed STOP" mouse lines are known for their high recombination efficiency, the recombination efficiency of conditional KO mouse lines depends on the design constraints imposed by the gene structure and the chromosomal location, cannot be predicted and needs to be experimentally determined. For an objective assessment of BG responsiveness to NE or ATP we considered a cell responsive if the calcium change trace exceeded 3x the standard deviation of the baseline signal for at least three consecutive imaging frames. With *Gli1-CreER^{T2}* driving recombination in the *Adra1acKO/cKO* mice, of 282 BG that were responsive to ATP 23 (8.2%) were not responsive to NE, while none (0%) of the 229 ATP-responsive wildtype BG were unresponsive to NE. With *Aldh111-CreER^{T2}* driving recombination in the *Adra1acKO/cKO* mice, of 308 BG that were responsive to ATP 30 (9.7%) were not responsive to NE, while none (0%) of the 214 ATP-responsive wildtype BG were unresponsive to NE. We have added illustrations of these proportions to Figure 4c lower left and Supplementary Figure 3b lower left. In addition, we are conducting in-situ hybridization experiments to confirm the low recombination efficiency of the *Adra1acKO/cKO* mouse line.

10. Not clear why the dura mater was removed following craniotomy as no drugs are applied locally.

Removal of the dura mater improves the optical conditions of chronic cranial windows and allows direct comparison to our previous work (PMID:24945771, PMID:28742117).

However, this manipulation does not affect vigilance-dependent astroglia calcium elevations (PMID:19447095). This manipulation is necessary to allow the local drug application experiments described below for the revision.

Reviewer #2:

First and foremost, while this is an elegant study using sophisticated methodology, the main conclusion is that EtOH increases locus coeruleus output, which excites Bergmann Glia. This is of course important information, but it is also one of many actions of EtOH in the brain, but in this manuscript there is no exploration of cellular mechanism or role in any relevant alcohol related trait/behavior that adds to our understanding of alcohol abuse and addiction. Thus, while this study adds to our already large list of cellular actions of EtOH, it does not substantially change our understanding of how such actions occur or how these specific actions relate to any aspects of alcohol related behaviors or clinical issues. That is of course fine, but it doesn't substantially advance the broad field. There are tens if not hundreds of reports detailing EtOH increasing or decreasing excitability of various neuronal or glial populations, and many of them have been explored enough to determine whether such actions are mediated pre or post synaptically, or both, and what specific mechanism mediate such responses, and how those responses affect behavioral actions of EtOH.

Specific comments.

1) Although my thoughts about writing style would normally be reserved for the minor comments section, in this case, I found the abstract and introduction, and related coverage of experimentation to be very confusing, generally, but especially in relation to what exactly is being studied. More specifically, in the abstract and introduction, the authors discuss/introduce astrocytic Ca²⁺ signaling in a very chaotic fashion, making it hard to understand what the authors think they are studying. The authors refer to/describe such signals and evoking behaviors as arousal, vigilance, sensory processing, locomotion-induced, adaptive regulation, attention, reward, motivation and stress, vigilant attention and cognitive performance. The authors then also describe how such glial signaling may be involved in development of AUD, brain development, adult function, synaptogenesis, pruning, ionic homeostasis, neurotransmitter transport, modulation of synaptic function, metabolic support and neurovascular coupling.

While many of these processes may be functionally related, overall this reads like a laundry list of astrocyte function without any guidance to how this leads to or relates to the ensuing elegant study.

While astrocytes, and their Ca²⁺ signaling are likely or clearly involved in these various functions (and many may be functionally linked), it is not clear how any are related to or drive the current studies of EtOH modulation of such signaling?

In parallel, there is no guidance or presentation of what is known about EtOH actions on astrocytes, and how the authors' experimentation will address any gaps. The authors can surely convey in 1-3 sentences that astrocytes do a lot more than we used to think they did, but should spend the rest of their writing leading us to the specifics of why they've focused on locus coeruleus, locomotion and their respective or interactive relationship to alcohol related behaviors.

We have rewritten our Introduction to provide a clearer logical stream that leads to the experiments. In short: We have previously demonstrated that locomotion-induced astroglia Ca^{2+} elevations are norepinephrine-dependent. Here, based on the expectation that any modulation of locus coeruleus activity might result in a consequential modulation of astroglia Ca^{2+} dynamics, we hypothesized that ethanol might have powerful effects on astroglia Ca^{2+} activation during awake behavior. The acute effect of ethanol on noradrenergic signaling in the awake brain is not clear. Now we discuss the variety of previously reported findings using different methodologies in more detail. All studies addressing the effect of ethanol on astrocyte Ca^{2+} dynamics have focused on cultured astrocytes, neglecting, as we now know, that the major driver of astrocyte Ca^{2+} activation in awake behaving mice is norepinephrine. Most studies found that acute ethanol increases astrocyte Ca^{2+} dynamics. We used a natural behavior as trigger of norepinephrine-dependent astroglia activation, avoiding artificially biasing manipulations such as electrical stimulation or opto-/chemogenetic approaches. By combining this behavioral approach with monitoring Ca^{2+} dynamics specifically in noradrenergic terminals and astroglia we found that acute ethanol does not enhance astroglia Ca^{2+} activation, but rather inhibits astroglia network activation through suppression of noradrenergic signaling.

2) In addition to the writing issues addressed above, it is actually important to consider experimentally what the recorded BG Ca^{2+} signals and modulation reflect behaviorally.

Important issues that should be addressed experimentally or at least with a coverage of clear evidence in the existing literature include:

A) Are such signals passive sensory evoked, motor output evoked, arousal evoked, vigilance evoked, or something else? Do such signals dictate behavior or are they responsive to behavior or both? And, whatever case can be made, the authors should be consistent in how they then refer to such signals.

We used a combination of global α_{1A} adrenergic receptor knockout and BG/astroglia-specific knockout to show that direct activation of α_{1A} adrenergic receptors on astroglia accounts for locomotion-induced Ca^{2+} signals. Therefore, any environmental or behavioral circumstance that can increase locus coeruleus activity is expected to increase astroglia Ca^{2+} activity. Exhaustive transsynaptic connectivity studies to establish the input/output connectome of locus coeruleus (PMID:26131933) have revealed extensive convergence of diverse inputs. These include diverse cortical areas, arousal controlling orexinergic afferents and even breathing control centers (PMID:28360327). Therefore, it is not surprising that very similar norepinephrine-dependent astroglia Ca^{2+} signals can be triggered by locomotion (voluntary as well as enforced), aversive stimuli such as air puffs to the face or tail, as well as sensory stimulation (e.g. whisker stimulation). From the reviewer's comments, we realize that we may have insufficiently defined our experimental paradigm. Accordingly, we have added information to the Introduction and consistently use now the term "locomotion-induced" whenever we merely describe the technical circumstance that caused astroglia Ca^{2+} signals and "vigilance" whenever we emphasize the behavioral state of the animal during physiological activation of locus coeruleus. We feel that it is important to use the term vigilance to appreciate that our findings are not restricted to locomotion.

B) Relatedly, why did the authors focus on "lobules simplex crus 1 of the hemispheres"? The case for how this subregion relates to what specific behavior the authors suggest they are studying should be made.

We picked one region of the cerebellum to avoid an additional source for variability. We have previously studied these BG signals in the cerebellar hemisphere (PMID:24945771) and these signals correspond to the locomotion-induced "flare" responses described for cerebellar vermis BG (PMID:19447095). In addition, the responses in cortical astrocytes suggest that astroglia can act as direct mediator of noradrenergic signaling wherever noradrenergic neurons project. Please note also our planned experiment in response to this reviewer's comment #5.

C) The authors should consider, explore and discuss how the above issues relate to the impact of EtOH generally, but in particular in the context of the blood concentration of alcohol that was likely achieved in the various studies. Alterations in BG signaling are first detected at 1mg/kg, and reach plateau at 1.5 and 2 mg/kg, which best estimation based on previous literature (Porcu et al. *ibid*) would be respectively 21, 30 and 42mM. What do such concentrations of EtOH do to rodent behavior? Certainly motor impairment, and at higher doses, loss of righting reflex. Does the associated change in BG Ca²⁺ signaling cause such motoric responses, or do alterations in such motoric responses cause the change in Ca²⁺ signaling? Given recent appreciation of cognitive/emotional roles of cerebellum (e.g. Carta et al. 2019 Science), are changes in BG signaling a reflection of reward, anxiety, aversion?

These questions are closely related to comment #2 of Reviewer #3. We are currently using the CatWalk XT (Noldus) approach to analyze the time course of gait performance following i.p. injections of 1 g/kg or 2 g/kg ethanol using the *Aldh1l1-CreER^{T2};Ai95* strain where we already have data about the recovery time course of the BG Ca²⁺ responsiveness following 2g/kg ethanol. Our preliminary observations suggest that motor coordination recovers much faster following 2 g/kg ethanol than BG Ca²⁺ responsiveness. These new data will provide experimental evidence that norepinephrine-dependent BG Ca²⁺ signals are not required for accurate motor function and also that accurate locomotion is not sufficient for BG Ca²⁺ signaling, rather they suggest a non-motor role for BG Ca²⁺ dynamics (at least over this acute time frame and our specific experimental paradigm) justifying further the term "vigilance-dependent".

3) While the authors touch on the issue of other cells possibly contributing to the altered signals, there are serious complications to consider if the signals truly only come from LC NE release. First and foremost, such high doses of EtOH are known to affect GABA and glutamate release within the molecular layer, both of which are known to affect BG Ca²⁺ signaling. Why aren't such signals and alteration by EtOH playing a role here? Relatedly, norepinephrine is known to modulate Purkinje cell excitability and it likely interacts with EtOH, how are all of these factors interacting?

Our combination of global alpha_{1A}-adrenergic receptor knockout (almost complete loss of locomotion-induced BG Ca²⁺ elevations) with conditional knockout in BG (complete loss of norepinephrine responsiveness in ~10% of BG and significant reduction of locomotion-induced BG Ca²⁺ elevations) suggests that direct noradrenergic signaling to BG is required and sufficient for global Ca²⁺ elevations seen in response to locomotion. This does not mean that the effects of GABA and glutamate on BG Ca²⁺, identified in studies using the acute slice preparation, do not exist in awake mice. It just suggests that these signals are either triggered by stimuli other than locomotion, or that they are more restricted to microdomains surrounding receptors and do not significantly contribute to the G_q-coupled receptor-mediated intracellular Ca²⁺ release that spreads throughout the entire BG, studied here. Similar conclusions have been arrived at independently by multiple groups – that astrocytes have receptors for many neurotransmitters (PMID:26814587), but exhibit Ca²⁺ transients *in vivo* during physiological conditions primarily in response to NE (PMID:24138901, PMID:24945771, PMID:25894291, PMID:27000523).

4) It would be informative to know whether actions on LC are mediated at the soma or on the terminals.

This is an excellent point. We are currently addressing this question through topical pharmacology experiments. Our plan is to first test if local application of 30 mM ethanol to the cerebellar cortex of a head-fixed awake mouse is sufficient to inhibit locomotion-induced BG Ca^{2+} elevations. If it is sufficient we will conclude that ethanol acts either at the soma as well as the terminals, or specifically at the terminals. If topical ethanol is not sufficient to inhibit BG we will conclude that ethanol acts at the LC somata. We will then inhibit BG Ca^{2+} elevations by i.p. injection of 2 g/kg ethanol and test if topical MK912 is sufficient to recover BG Ca^{2+} signals. Please note, the more direct approach would be to monitor LC terminal Ca^{2+} dynamics for this experiment; however, while we may attempt this, the BG Ca^{2+} signals have a much higher signal/noise and we have already demonstrated in acute slices that 30 mM ethanol has no significant effect on the norepinephrine responsiveness of BG.

Our preliminary data pursuing this approach indicate that the ethanol site of action is at the soma while MK912 action at the terminals is sufficient to restore NE release.

5) The behaviorally associated BG Ca^{2+} signals are surprisingly widespread and uniform. Are similar responses seen in other regions of the cerebellum, are they or can they be evoked by unrelated behaviors, and are such responses similarly modulated by EtOH or modulation of adrenergic receptors?

We are experimentally addressing this question. From our findings that ethanol also inhibits locomotion-induced primary visual cortex astrocyte Ca^{2+} elevations, that locomotion also induces Ca^{2+} elevations in cerebellar vermis BG (PMID:19447095) and that our slice experiments for evaluation of conditional α_{1A} -adrenergic receptor KO covered the hemisphere as well as vermis, we predict that locomotion induces Ca^{2+} elevations in vermal BG that are inhibited by 2 g/kg i.p. ethanol. We further predict that these responses can also be evoked by air puff application to the face of the mouse and that ethanol also inhibits air puff-evoked responses.

6) Given the clear ability to pharmacologically (or genetically: the BG specific α_1 KOs!) prevent EtOH suppression of BG Ca^{2+} signals, it would add enormous excitement to this manuscript, to determine if doing so affected some EtOH related behavior (especially consumption and/or motor impairment), which would powerfully link BG responses and the cerebellum to EtOH behavioral actions.

We are currently conducting experiments described in response to comment 2C above. Preliminary studies suggest that there is a dissociation between the recovery of BG Ca^{2+} signaling and motor coordination. These data will provide experimental evidence that locomotion-induced BG Ca^{2+} elevations may not be required for accurate motor coordination, rather for non-motor functions of the cerebellum. We suspect that norepinephrine-dependent BG Ca^{2+} dynamics support attention requiring efforts; however, more refined behavioral assessments will need to be developed to quantify attention in mice and test this hypothesis in future studies. We will add additional discussion of this point to the manuscript.

Reviewer #3:

Major Comments:

1) The evidence for involvement of α_1 adrenergic receptors in the locomotion-related BG calcium increases comes mainly from the experiments with knockout mice. However, as the authors indicate this knockout does not appear to be complete. Previous work from this group

indicated involvement of NE and alpha1 receptors in these responses, but a nonspecific antagonist trazodone was used. Why not repeat these experiments with a selective alpha1 antagonist?

We have added Supplementary Figure 1 that demonstrates that the alpha1 adrenergic receptor-specific antagonist prazosin inhibits locomotion-induced BG Ca²⁺ elevations, comparable to the global knockout of just alpha_{1A} adrenergic receptors in astroglia (Figure 3).

2) While it is clear that the ethanol treatment has a minimal and short-lasting effect on actual locomotor performance, it would be good to know how the other genetic and pharmacological manipulations affected movement on the treadmill. Ultimately, it is important to determine if glial calcium signaling is involved in locomotor function, or provides a general signal for attention or arousal. Obviously, this issue can't be resolved in the present study, but the author should discuss the subject.

We are currently using the CatWalk XT (Noldus) approach to analyze the time course of gait performance following i.p. injections of 1 g/kg or 2 g/kg ethanol using the *Aldh1l1-CreER^{T2};Ai95* strain where we already have data about the recovery time course of the BG Ca²⁺ responsiveness following 2g/kg ethanol. Our preliminary observations suggest that the motor coordination recovers much faster following 2 g/kg ethanol than BG Ca²⁺ responsiveness. These new data will provide experimental evidence that norepinephrine-dependent BG Ca²⁺ signals are not required for accurate motor function and also that accurate locomotion is not sufficient for BG Ca²⁺ signaling, rather they will suggest a non-motor role for BG Ca²⁺ dynamics. We will also analyze gait performance in global alpha_{1A} adrenergic KO mice.

3) The size of the ROIs shown in figure indicate that the authors weren't attempting to measure calcium changes at the single bouton level. Thus, it should be made clear to the reader that fields of axons and terminals were imaged in this experiment.

We have added this information where we introduce these experiments.

4) While the LC provides a strong NE projection to cerebellum, recent evidence suggests that other noradrenergic nuclei also project there (e.g. Robertson et al., Nature Neuroscience, 2013). Thus, the authors may want to be more inclusive in discussing which nuclei might contribute to the NE effects.

We appreciate that there are non-LC sources of NE to the cerebellum and will refer to this finding in the Discussion.

Minor Comments:

i. Figure 6C, it's not clear that the lines connecting the points serve any purpose. The color coding of the symbols is sufficient to convey the story. Also, the light gray symbols are hard to see against a white background, so another color should be used for the 0.5 g/kg group.

We eliminated the connecting lines (in Figures 1C and 6C) and added a colored background to be able to preserve the grayscale representation of ethanol groups.

ii. It appears that the deltaF/F values were used to detect voluntary locomotion-induced calcium increases, which makes sense. However, was this corroborated by the EMG data that was also collected?

We collected the EMG signal to monitor whether mice that were exposed to ethanol were still active on the treadmill. For quantification of Ca²⁺ responses we were most concerned about voluntary activity events that were intense enough to cause a detectable Ca²⁺ rise and therefore could depress the consecutive response to enforced locomotion. Thus, we included trials with EMG activity increases for analysis as long as no Ca²⁺ rise could be detected during the respective baseline.

iii. Page 32, the authors mention the use of a Goat Anti-Mouse IgG1, but this reviewer did not see any mouse primary antibodies in the methods description.

We have now added the missing information that the S100 β antibody was a mouse (IgG₁) anti-S100 β antibody.

iv. Page 38, line 855, the statement "...suggesting that noradrenergic receptors were knocked down to a functionally relevant extent." is not accurate, as the loss of signal is not necessarily due to loss of receptors, but may also include impaired signaling downstream of the receptor.

We have added a reference to the main text and a brief note proceeding previous page 38, line 855 to explain that BG ATP-induced Ca²⁺ rises are dependent on intracellular Ca²⁺ release and mediated by Gq-coupled receptors (P2Y2 and/or P2Y4). Therefore, any impaired signaling downstream of the Gq protein would have been expected to affect responses to NE and ATP similarly.

v. There are several grammar and wording issues that should be corrected, particularly in the methods section.

We have done best efforts to eliminate grammar and wording issues.

REVIEWER COMMENTS

Reviewer #1 (Remarks to the Author):

This manuscript is much improved with the addition of new analysis and experiments as well as the substantial rewrites.

However, the authors' main conclusion is that $\alpha 1A$ -adrenergic receptors on astroglia is required for their global activation during enhanced vigilance based on the conditional $\alpha 1A$ adrenergic receptor knockout mice. Although on the genetic level they have a cKO, functionally they appear not to, as the vast majority of cells continue to respond to NE stimulation. Therefore, this major conclusion is not well supported.

The description of the conditional KO results is somewhat misleading (L.357 "...reduced those responses considerably"; L.360 "loss of BG responsiveness to NE was incomplete"). I think a reduction of 8% with a KO does not support that language. "Somewhat reduced" is more accurate. In the abstract it says "selective abolition" of astroglia Ca^{2+} . Again 8-9% fewer cells responding to NE is hardly abolition of astroglia Ca^{2+} .

Figure 4c left inset- the data should be presented as proportion of cells rather than number of cells as the larger sample in the KO makes the difference appear larger.

Figure 10 needs to be better labeled and described in the legend. It is not clear which mice are described in 10c and 10d.

Reviewer #2 (Remarks to the Author):

I would like to start off by apologizing to the authors for my slowness in providing my review/feedback on this resubmission. I am sincerely sorry for my slow progress on this, but I wanted to assure you it was not out of any form of malfeasance, just dealing with the new reality of life. And while reviewing a manuscript can obviously be done safely at home or in one's office, so isn't inherently affected by CV-19, every other aspect of my job obligations and home life have been, especially with students arriving, classes starting and associated cases of CV-19 rising. And, I have my own child, and in person class is not happening, so we've been scrambling to figure out what we're going to do. Not meant to be excuses, but just an explanation of sorts, and a sincere apology.

With that aside, this revised manuscript presents experiments designed to improve our understanding of how alcohol (EtOH) affects locus coeruleus driven, adrenergic modulation of Bergmann glial cell astrocytes in the cerebellar cortex. The authors use multiphoton confocal fluorescence imaging of gCaMP (a genetically encoded Ca^{2+} sensitive fluorophore), in awake, but head fixed mice on a treadmill, to assay EtOH modulation of locomotion associated Bergmann glial (BG) Ca^{2+} signaling. Please see comments on the original submission for specific details.

The authors have made a sincere effort to address many of the concerns expressed by me and the other reviewers.

Of most relevance, they have conducted numerous additional experiments to nail down the site of action of EtOH for the originally reported suppression of motor-triggered Bergmann glia (BG) Ca^{2+} elevations. In so doing the authors have now made a convincing case that the suppression of such Ca^{2+} signals are likely via actions on Locus Coeruleus (LC) neuronal cell bodies or dendrites, given that EtOH does not affect BG responses to exogenous NE in acute cerebellar slices, local application of

EtOH in vivo does not affect BG Ca²⁺ responses, but local application of an α_2 autoreceptor (which increases NE release) helps recovery of systemic EtOH suppression of BG Ca²⁺ responses. And, motor induced increased NE terminal Ca²⁺ responses correlates with BG Ca²⁺ responses in vivo, during suppression by EtOH and in various other pharmacological scenarios.

Second, the authors did a nice job of rewriting the introduction to better frame this project in the relevant contexts, and they have also provided blood alcohol concentrations to relate to most of the outcomes and time courses of most of the reported experiments.

The authors have, along with revamping the introduction, language in methods and discussion, made a more clear case for why their in vivo study in awake animals actually adds substantially to our understanding of how EtOH affects the LC to astrocyte Ca²⁺ signaling cascade, without concerns about potential slice, culture or anesthesia related artifacts.

Based on these improvements, the specific underlying mechanisms of EtOH suppression of motor induced glial Ca²⁺ signaling have now been more clearly delineated, and overall the relevance of these mechanistic findings have been more clearly related to existing literature and conceptual frameworks. Consequently, I think the quality of the design and interpretation, and the context in which it is now presented all warrant publication.

My only remaining reservation is that it is still unclear what the observed signals do, and how their modulation by EtOH affects behavioral responses to EtOH and/or how this relates to alcohol use and abuse? Importantly, while it is now very clear that the cerebellum is critically involved in reward, emotion and cognition, and it is arguably uniquely sensitive to EtOH, it is also clear that the cerebellum and its response to EtOH plays a dominant role in EtOH induced ataxia/motor incoordination, neither of which is influenced by the reported EtOH suppression of LC driven, NE mediated BG Ca²⁺ signal (Fig. 10). Importantly in this context, while motor impairment is an easily quantified and cerebellar constrained readout of EtOH action in the cerebellum, presumably other, non-motor roles of the cerebellum are similarly disrupted in parallel. So, if the BG Ca²⁺ are not involved in the motor impairing effects of EtOH, what evidence is there that it is involved in other aspects of cerebellar processing?

Moreover, the BG Ca²⁺ signals are primarily suppressed only by fairly high concentrations of EtOH (2g/kg injections leading to 30-40 mM BECs), which existing literature has shown affects almost every synapse in the cerebellum (Including multiple studies by Dunwiddie's group, Valenzuela's group, Dar's, group, Hansel's group, Rossi's group, Otis'/Olsen's group). Relatedly, although alcohol abusive patients may regularly achieve such BECs, few if any non-human models (including the rodent lines used in this publication) will voluntarily consume to that level of intoxication, with the range typically being about 0 to 25mM. So, while it is possible that as the authors imply, that the BG Ca²⁺ response has some role in cognitive/emotional responses to EtOH (which would be fascinatingly important to establish), based on the data in this manuscript, it seems most likely that the observed responses are simply one of many impairments of brain signaling that occur during extreme inebriation, and as of yet has no obvious functional/behavioral ramification, much less implication for addressing the clinical situation. That is of course fine, and basic research is critically important, but this type of information seems more relevant to a more focused journal.

At this stage, I think the science is now sound and definitive in terms of mechanism, and the presentation is clear and contextually placed. So, it seems to be up to the publishers/editors whether they feel this level of broad advance meets their standards. The science is sound and clear, its just not abundantly clear how this affects our broad thinking about alcohol use and abuse. In short, while the underlying mechanisms are now clear, it seems unlikely that BG Ca²⁺ signals influence EtOH consumption or even EtOH related impairment, but are rather just one of many neuronal processes that malfunction during severe inebriation, despite having little impact on overall behavioral outcome.

Reviewer #3 (Remarks to the Author):

The authors have improved the manuscript with extensive revisions, and clarified several points. Overall, the findings are now clear. While it is unfortunate that the astrocytic calcium signals could not yet be related to a particular behavior, the paper does provide strong evidence of the alcohol effects on these responses and the mechanisms underlying these effects. There are a couple of revisions that might be helpful:

1. The authors might consider the possibility of compensation by another alpha1 receptor subtype in discussing the lack complete loss of response in the BG Cre-dependent knockout experiment, although it is unclear why this wouldn't happen in the global knockout mice.
2. Line 310, it might be better to say alcohol intoxication rather than toxicity.

We would like to thank the reviewers for their constructive comments and suggestions. Reviewer #1's comments have prompted us to slightly edit Figs. 4, 10 and Supplementary Fig. 4. Reviewers #1 and #2, in addition, have prompted us to edit the manuscript text at the locations indicated below and highlighted in blue in the text document. We think that these refinements will help improve the clarity of the manuscript.

Point-By-Point Response to Reviewers' Comments:

Reviewer #1:

This manuscript is much improved with the addition of new analysis and experiments as well as the substantial rewrites.

However, the authors' main conclusion is that $\alpha 1A$ -adrenergic receptors on astroglia is required for their global activation during enhanced vigilance based on the conditional $\alpha 1A$ adrenergic receptor knockout mice. Although on the genetic level they have a cKO, functionally they appear not to, as the vast majority of cells continue to respond to NE stimulation. Therefore, this major conclusion is not well supported.

The description of the conditional KO results is somewhat misleading (L.357 "...reduced those responses considerably"; L.360 "loss of BG responsiveness to NE was incomplete"). I think a reduction of 8% with a KO does not support that language. "Somewhat reduced" is more accurate. In the abstract it says "selective abolition" of astroglia Ca^{2+} . Again 8-9% fewer cells responding to NE is hardly abolition of astroglia Ca^{2+} .

The term "reduced those responses considerably" in line 357 (now 360) refers to the population *in vivo* data in Fig. 4b and Supplementary Fig. 4a, which show that BG activation was reduced by 37% (BG-specific KO) and 43% (astroglia-specific KO), respectively. The 8% reduction refers to the proportion of BG that completely lost responsiveness to NE, based on an arbitrarily set threshold. To avoid confusion regarding data interpretation, we have now revised the statement to "reduced those responses considerably, but incompletely (line 360)" in one sentence, similar to how we describe these findings in the Results section (line 182).

The "selective abolition" in the abstract refers to the study of motor coordination in global $\alpha 1A$ -adrenergic receptor knockout mice presented in Fig. 10e. We think that Fig. 3 justifies to use the term "abolition", and the ~94% loss of *Adra1a* mRNA in total cerebellum tissue from recombination efficiency-dependent, astroglia-selective gene deletion mice (Supplementary Fig. 5) justifies the attribute "selective". However, since from a technical perspective we used global knockout mice, we appreciate the reviewer's concern that this could be misunderstood and decided to remove the attribute "selective".

Figure 4c left inset- the data should be presented as proportion of cells rather than number of cells as the larger sample in the KO makes the difference appear larger.

We thank the reviewer for suggesting that we emphasize proportions within BG populations rather than absolute cell counts. Following this advice, we replaced the bar graphs in Fig. 4c left inset and Supplementary Fig. 4b left inset with cumulative probability distribution plots. The advantage of this method of data representation is that the reader gains unbiased access to understanding the effect of BG- or astroglia-specific KO of α 1A-adrenergic receptors without data selection by any arbitrarily set threshold. In the Results section (lines 194 - 199), we have added numbers related to the proportion of BG that responded equally to NE and ATP to guide the reader with one example of how to use the cumulative probability distribution plot.

Figure 10 needs to be better labeled and described in the legend. It is not clear which mice are described in 10c and 10d.

We have labeled figure panels 10c and 10d as well as have mentioned the mouse strain in the legend (line 1217).

Reviewer #2:

With that aside, this revised manuscript presents experiments designed to improve our understanding of how alcohol (EtOH) affects locus coeruleus driven, adrenergic modulation of Bergmann glial cell astrocytes in the cerebellar cortex. The authors use multiphoton confocal fluorescence imaging of gCaMP (a genetically encoded Ca²⁺ sensitive fluorophore), in awake, but head fixed mice on a treadmill, to assay EtOH modulation of locomotion associated Bergmann glial (BG) Ca²⁺ signaling. Please see comments on the original submission for specific details.

The authors have made a sincere effort to address many of the concerns expressed by me and the other reviewers.

Of most relevance, they have conducted numerous additional experiments to nail down the site of action of EtOH for the originally reported suppression of motor-triggered Bergmann glia (BG) Ca²⁺ elevations. In so doing the authors have now made a convincing case that the suppression of such Ca²⁺ signals are likely via actions on Locus Coeruleus (LC) neuronal cell bodies or dendrites, given that EtOH does not affect BG responses to exogenous NE in acute cerebellar slices, local application of EtOH in vivo does not affect BG Ca²⁺ responses, but local application of an α 2 autoreceptor (which increases NE release) helps recovery of systemic EtOH suppression of BG Ca²⁺ responses. And, motor induced increased NE terminal Ca²⁺ responses correlates with BG Ca²⁺ responses in vivo, during suppression by EtOH and in various other pharmacological scenarios.

Second, the authors did a nice job of rewriting the introduction to better frame this project in the relevant contexts, and they have also provided blood alcohol concentrations to relate to most of the outcomes and time courses of most of the reported experiments.

The authors have, along with revamping the introduction, language in methods and discussion, made a more clear case for why their in vivo study in awake animals actually adds substantially to our understanding of how EtOH affects the LC to astrocyte Ca²⁺ signaling cascade, without concerns about potential slice, culture or anesthesia related artifacts.

Based on these improvements, the specific underlying mechanisms of EtOH suppression of motor induced glial Ca²⁺ signaling have now been more clearly delineated, and overall the relevance of these mechanistic findings have been more clearly related to existing literature and conceptual frameworks. Consequently, I think the quality of the design and interpretation, and the context in which it is now presented all warrant publication.

We would like to thank the reviewer for the appreciation of our efforts.

My only remaining reservation is that it is still unclear what the observed signals do, and how their modulation by EtOH affects behavioral responses to EtOH and/or how this relates to alcohol use and abuse? Importantly, while it is now very clear that the cerebellum is critically involved in reward, emotion and cognition, and it is arguably uniquely sensitive to EtOH, it is also clear that the cerebellum and its response to EtOH plays a dominant role in EtOH induced ataxia/motor incoordination, neither of which is influenced by the reported EtOH suppression of LC driven, NE mediated BG Ca²⁺ signal (Fig. 10). Importantly in this context, while motor impairment is an easily quantified and cerebellar constrained readout of EtOH action in the cerebellum, presumably other, non-motor roles of the cerebellum are similarly disrupted in parallel. So, if the BG Ca²⁺ are not involved in the motor impairing effects of EtOH, what evidence is there that it is involved in other aspects of cerebellar processing?

This is indeed an important question, motivated by the current findings, that will need to be addressed in subsequent studies designed to assess cognitive performance in mice.

Moreover, the BG Ca²⁺ signals are primarily suppressed only by fairly high concentrations of EtOH (2g/kg injections leading to 30-40 mM BECs), which existing literature has shown affects almost every synapse in the cerebellum (Including multiple studies by Dunwiddie's group, Valenzuela's group, Dar's, group, Hansel's group, Rossi's group, Otis/Olsen's group). Relatedly, although alcohol abusive patients may regularly achieve such BECs, few if any non-human models (including the rodent lines used in this publication) will voluntarily consume to that level of intoxication, with the range typically being about 0 to 25mM. So, while it is possible that as the authors imply, that

the BG Ca²⁺ response has some role in cognitive/emotional responses to EtOH (which would be fascinatingly important to establish), based on the data in this manuscript, it seems most likely that the observed responses are simply one of many impairments of brain signaling that occur during extreme inebriation, and as of yet has no obvious functional/behavioral ramification, much less implication for addressing the clinical situation. That is of course fine, and basic research is critically important, but this type of information seems more relevant to a more focused journal.

We appreciate the Reviewer's concerns about the use of rodent models and the pleotropic effects of ethanol. Nevertheless, in order to understand the mechanisms responsible for the acute effects of ethanol intoxication, it is necessary to define these different sites of actions and their consequences. Here, we examined the inhibition of vigilance-dependent astroglia Ca²⁺ activation in awake behaving mice – this is the first time that such an analysis has been performed. Our results indicate that even the highest dosage of ethanol used, 2 g/kg i.p., impaired motor coordination only temporarily. Importantly, BG activation was still impaired 45 min after EtOH injection (Fig. 2a to c), when the mice performed normally in the motor coordination task (Fig. 10c). It further needs to be emphasized that 1.5 g/kg i.p. ethanol was already sufficient to inhibit BG activation (Fig. 1f), suggesting that impairment of astroglial activation will occur with much lower ethanol exposure and outlast motor deficits.

At this stage, I think the science is now sound and definitive in terms of mechanism, and the presentation is clear and contextually placed. So, it seems to be up to the publishers/editors whether they feel this level of broad advance meets their standards. The science is sound and clear, its just not abundantly clear how this affects our broad thinking about alcohol use and abuse. In short, while the underlying mechanisms are now clear, it seems unlikely that BG Ca²⁺ signals influence EtOH consumption or even EtOH related impairment, but are rather just one of many neuronal processes that malfunction during severe inebriation, despite having little impact on overall behavioral outcome.

Thank you for these comments.

Reviewer #3:

The authors have improved the manuscript with extensive revisions, and clarified several points. Overall, the findings are now clear. While it is unfortunate that the astrocytic calcium signals could not yet be related to a particular behavior, the paper does provide strong evidence of the alcohol effects on these responses and the mechanisms underlying these effects

There are a couple of revisions that might be helpful:

1. *The authors might consider the possibility of compensation by another alpha1 receptor subtype in discussing the lack complete loss of response in the BG Cre-dependent knockout experiment, although it is unclear why this wouldn't happen in the*

global knockout mice.

We have added this possibility to the discussion (lines 364 - 366).

2. Line 310, it might be better to say alcohol intoxication rather than toxicity.

We replaced the word "toxicity" in line 311 with "intoxication".

REVIEWERS' COMMENTS

Reviewer #1 (Remarks to the Author):

The authors addressed the remaining concerns.

Point-By-Point Response to Reviewers' Comments:

Reviewer #1:

The authors addressed the remaining concerns.

We would like to thank reviewer #1 that he/she considers all remaining concerns addressed.